# Gasdermin E dictates inflammatory responses by controlling the mode of neutrophil death

Fengxia Ma [1,2,5] ✉, Laxman Ghimire [3,5], Qian Ren [1,2,5], Yuping Fan[1,2], Tong Chen [1,2], Arumugam Balasubramanian [3], Alan Hsu [3], Fei Liu[3], Hongbo Yu[4], Xuemei Xie[3], Rong Xu[3] & Hongbo R. Luo [3] ✉

Both lytic and apoptotic cell death remove senescent and damaged cells in living organisms. However, they elicit contrasting pro- and anti-inflammatory responses, respectively. The precise cellular mechanism that governs the choice between these two modes of death remains incompletely understood. Here we identify Gasdermin E (GSDME) as a master switch for neutrophil lytic pyroptotic death. The tightly regulated GSDME cleavage and activation in aging neutrophils are mediated by proteinase-3 and caspase-3, leading to pyroptosis. GSDME deficiency does not alter neutrophil overall survival rate; instead, it specifically precludes pyroptosis and skews neutrophil death towards apoptosis, thereby attenuating inflammatory responses due to augmented efferocytosis of apoptotic neutrophils by macrophages. In a clinically relevant acid-aspiration-induced lung injury model, neutrophil-specific deletion of GSDME reduces pulmonary inflammation, facilitates inflammation resolution, and alleviates lung injury. Thus, by controlling the mode of neutrophil death, GSDME dictates host inflammatory outcomes, providing a potential therapeutic target for infectious and inflammatory diseases.

Neutrophils - terminally differentiated cells with a very short life-span (7–20 h) - are continuously generated in the bone marrow (BM) through tightly regulated granulopoiesis. About $10^{11}$ neutrophils are produced every day and a similar number undergo programmed cell death (PCD), keeping their number constant under normal conditions[1–5]. Neutrophil PCD is also critical for modulating neutrophil number and accumulation at sites of infection and inflammation. Neutrophils die even in the absence of extracellular stimuli and independent of pathogen invasion, which is therefore known as constitutive or spontaneous neutrophil death[6–8]. It is well documented that neutrophil death is a key component in innate immune regulation ([8] and references therein).

PCD, which can be lytic or non-lytic, describes a tightly regulated form of cellular suicide essential in many pathophysiological processes[9]. Apoptosis is a non-lytic and anti-inflammatory PCD, as cellular components are sequestered into apoptotic bodies that prevent the release of proinflammatory constituents. Necroptosis, pyroptosis and PANoptosis are lytic forms of PCD that release pro-inflammatory substrates such as damage-associated molecular patterns (DAMPs), ultimately amplifying inflammation[10–12]. Neutrophils

[1] State Key Laboratory of Experimental Hematology, National Clinical Research Center for Blood Diseases, Haihe Laboratory of Cell Ecosystem, CAMS Key Laboratory for Prevention and Control of Hematological Disease Treatment Related Infection, Institute of Hematology & Blood Diseases Hospital, Chinese Academy of Medical Sciences & Peking Union Medical College, Tianjin, China. [2] Tianjin Institutes of Health Science, Chinese Academy of Medical Sciences, Tianjin, China. [3]Department of Pathology, Dana-Farber/Harvard Cancer Center, PhD Program in Immunology, Harvard Medical School; Department of Laboratory Medicine, Boston Children's Hospital, Enders Research Building, Room 811, Boston, MA 02115, USA. [4]VA Boston Healthcare System, Department of Pathology and Laboratory Medicine, 1400 VFW Parkway, West Roxbury, MA 02132, USA. [5]These authors contributed equally: Fengxia Ma, Laxman Ghimire, Qian Ren. ✉e-mail: mafengxia@ihcams.ac.cn; Hongbo.Luo@childrens.harvard.edu

die by both anti-inflammatory apoptosis and pro-inflammatory lytic cell death[8,10,13–17]. During inflammation, apoptotic neutrophils are quickly recognized, engulfed, and cleared by macrophages via a highly regulated process known as efferocytosis, which triggers anti-inflammatory cytokine production that facilitates the resolution of inflammation[18–22]. While this timely clearance of apoptotic neutrophils prevents a cascade of further inflammation, it also helps to clear pathogens that have been engulfed but not cleared by these neutrophils[23,24]. Neutrophil PCD is therefore heterogeneous, but the underlying mechanisms dictating the type of cell death remain uncertain, even though understanding these mechanisms is important for resolving or controlling inflammation in inflammatory diseases.

Gasdermins are a conserved protein family with six paralogues identified in humans (GSDMA, GSDMB, GSDMC, GSDMD, GSDME, and DFNB59) and ten paralogues identified in mice (GSDMA1-3, GSDMC1-4, GSDMD, GSDME, and DFNB59)[25–27]. All gasdermins (except DFNB59) share the common structure of a cytotoxic N-terminus and an inhibitory C terminus connected by a linker. Gasdermins are cleaved by proteinases to generate the active N-terminus domain, which forms pores that rupture the plasma membrane and cause lytic pyroptotic cell death[28–33]. Here we identified GSDME as the major regulator determining the mode of neutrophil death. In aging neutrophils, GSDME was cleaved and activated, leading to lytic neutrophil death. Interestingly, while GSDME disruption did not alter the number of healthy neutrophils, it completely abolished neutrophil pyroptotic death and skewed neutrophil death to apoptosis both in vitro and in vivo at sites of inflammation. Elevated neutrophil apoptosis enhanced their efferocytotic clearance by macrophages, augmenting anti-inflammatory responses in murine peritonitis and lipopoly-saccharide (LPS)-induced acute lung injury. Importantly, in a clinically relevant acid aspiration–induced lung injury model, neutrophil-specific GSDME disruption significantly alleviated host inflammatory responses and consequently attenuated lung injury. Collectively, we reveal a neutrophil-centered mechanism for modulating host inflammatory responses through control of the mode of neutrophil death and demonstrate its therapeutic potential. Neutrophil GSDME regulates the mode of cell death, which in turn dictates the host inflammatory response. GSDME disruption can switch caspase-3-mediated pyroptosis to caspase-3-mediated apoptosis. By manipulating neutrophil GSDME, the host response can be modulated to improve outcomes in inflammatory diseases.

## Results

### GSDME cleavage and activation are tightly regulated in aging neutrophils and disruption of GSDME skews neutrophil death to apoptosis

An analysis of mouse scRNA-seq data revealed that, besides *Gsdmd*, both immature and mature neutrophils expressed high levels of *Gsdme* (*Dfna5*) (Fig. S1). We therefore wondered whether GSDME also plays a role in neutrophil death and host defenses. First, we isolated bone marrow (BM)-derived neutrophils from wild type (WT) mice and cultured them for 16 h. GSDME was gradually cleaved during spontaneous neutrophil death, suggesting a tight regulation of its activity in this process (Fig. 1a). To further investigate the role of GSDME in neutrophil death, we generated a *Gsdme* knockout (KO) mouse (Fig. S2a and S2b), and total and differential leukocyte counts in the peripheral blood and BM were unaffected by *Gsdme* KO (Fig. S2c and S2d). We first assessed death of neutrophils purified from the BM of WT and *Gsdme* KO mice cultured up to 24 h by fluorescence microscopy and, consistent with our previous report[13], observed heterogeneous neutrophil spontaneous death (Fig. 1b and Movie S1 & S2). Besides shrinking apoptotic cells, some WT neutrophils were swollen or "puffed"[13](Fig. 1b and Movie S1 & S2), with their number peaking at 12 h (Fig. 1c, d). Swollen cells were propidium iodide (PI) positive, suggesting a lytic form of cell death[34] (Fig. 1b). Strikingly, these swollen

cells were nearly completely suppressed by *Gsdme* KO, even at 24 h, indicating abolition of lytic cell death after GSDME deletion (Fig. 1c, d). By contrast, the percentage and number of apoptotic annexin V (AV)-positive cells increased significantly in KO neutrophils compared with WT (Fig. 1d). Of note, both the swollen and shrinking cell populations contained PI-positive cells (Fig. 1c, d). Morphologically, all PI-positive neutrophils from KO mice were smaller but with stronger fluorescence intensity compared with swollen PI-positive cells in the WT neutrophil population (Fig. 1c), suggestive of apoptosis. In the WT population, most PI-positive cells detected at early time points (e.g., 8 and 12 h) were swollen neutrophils undergoing lytic death, and most PI-positive cells detected at late time points (e.g., 20 and 24 h) were derived from apoptotic neutrophils (Fig. 1b–d). To mitigate the impact of continuous culturing under a microscope on morphological analysis, we also conducted morphological examinations at fixed time points without continuous monitoring. In addition, we used another cell-impermeant fluorogenic nucleic acid dye, SYTOX Orange, to stain dying neutrophils. Essentially identical results were observed, with GSDME-deficient neutrophils exhibiting increased apoptosis but markedly decreased lytic death (Fig. S3).

By tracking each cell in time-lapse videos, their mode of death was determined based on both morphology and AV/PI staining (Fig. 1b). Significantly more neutrophils were recovered over time from the KO population than the WT population (Fig. 1e), and disappearing cells were mainly swollen and ultimately lysed in culture. There were significantly fewer swollen, lysed cells in the KO population, consistent with inhibition of lytic cell death (Fig. 1e). Of note, because many dead neutrophils disappear (turning into debris) during the process, the most accurate measurement for neutrophil death is the absolute number of neutrophils at each time point, rather than the percentage. Although the percentages of healthy (PI/AV-negative) cells in the *Gsdme* KO group were lower compared with WT, there were equal numbers of healthy cells at each time point in each group (Fig. 1f–g), suggesting that GSDME disruption only altered the mode of neutrophil death (i.e., lytic to apoptotic) but not the overall survival rate (Fig. 1g), with the proportion of cells undergoing apoptosis increasing significantly in the KO population (Fig. 1g). We also measured the levels of lactate dehydrogenase activity (LDH), a biomarker of lytic cell death. *Gsdme* KO neutrophils exhibited a significant reduction in LDH release during their aging process, further suggesting a decrease in lytic death compared to their WT counterparts (Fig. 1h).

To further investigate the role of GSDME in neutrophil death, bone marrow neutrophils from WT and *Gsdme* KO mice were cultured for different times, stained, and analyzed by flow cytometry (Fig. 2a). A large proportion of swollen or lytic cells detected by microscopy (Fig. 1b, c) in the WT group were lost while pipetting and processing for flow cytometry[13]. Thus, to accurately measure absolute numbers of cell populations at each time point, flow cytometry beads were included in each FACS sample (Fig. S4). Cell death analyzed by flow cytometry was largely consistent with that seen by microscopy. Although the percentage of healthy neutrophils (AV/PI double-negative) was lower in the *Gsdme* KO group than the WT group (Fig. 2a, b), the absolute number of healthy cells was not affected by *Gsdme* knockout (Fig. 2c). Similarly, the number of apoptotic AV⁺PI⁻ cells was higher in KO than WT neutrophils (Fig. 2c). Interestingly, the granularity and size (assessed by side (SSC) and forward (FSC) scatter) of WT and KO neutrophils were different (Fig. 2d). The FSC of the P2 subpopulation (mostly single AV positive) from WT neutrophils was much lower than the FSC of the P1 population in the same WT population (Fig. 2d). However, the SSC and FSC of the P1 and P2 subpopulations in KO neutrophils were similar. The proportions of AV and PI positive cells in each subpopulation were also different between WT and KO (Fig. 2d).

Consistent with the higher proportion of apoptotic cells, cleaved caspase-3 was higher in *Gsdme* KO neutrophils at each time point (Fig. 2e). Similarly, cleavage of PARP1, a nuclear poly (ADP-ribose)

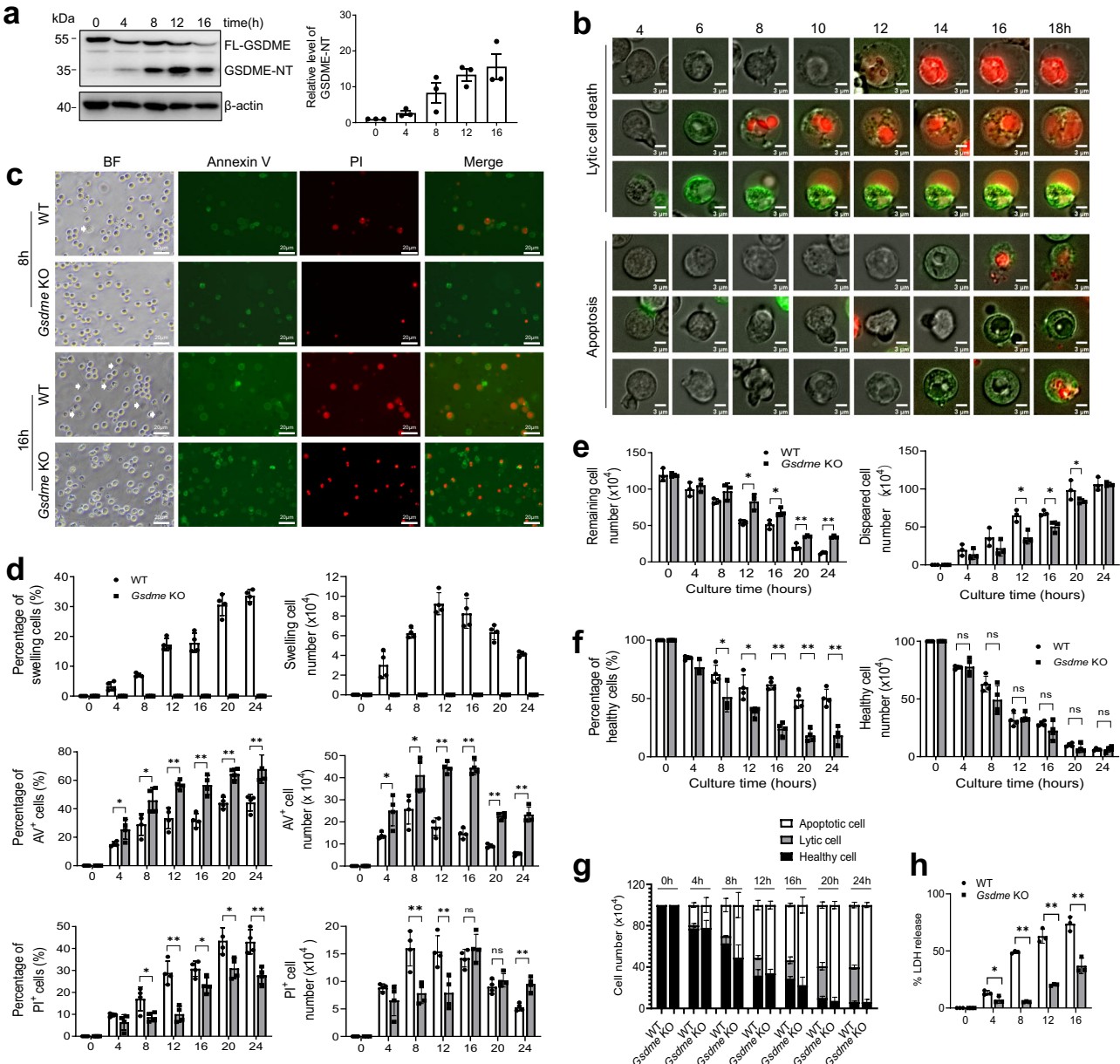

**Fig. 1 | GSDME disruption abolished neutrophil lytic death without affecting overall neutrophil survival. a** Cleavage of GSDME during neutrophil spontaneous death. Bone marrow-derived neutrophils from WT mice were used. Results are representative of at least three biological replicates. All data are represented as mean ± SD, $n = 3$ independent repeats. **b** Representative images of neutrophils undergoing apoptosis and lytic cell death. Results are representative of at least three independent experiments. **c** Representative images of aging neutrophils at the indicated time points. The white arrow heads indicate swollen or "puffed" cells (lytic cell death). PI propidium iodide, BF bright field. Results are representative of at least three independent experiments. **d** The percentage and number of swollen, AV-positive, and PI-positive cells. All data are represented as mean ± SD. $n = 4$ independent repeats, *$P < 0.05$, **$P < 0.01$, ns non-significant (Percentage of AV+ cells: 4 h, $P = 0.0273$; 8 h, $P = 0.0285$; 12 h, $P = 0.0011$; 16 h, $P = 0.001$; 20 h, $P = 0.0004$; 24 h, $P = 0.0067$. Percentage of PI+ cells: 8 h, $P = 0.0297$; 12 h, $P = 0.0011$; 16 h, $P = 0.022$; 20 h, $P = 0.0168$; 24 h, $P = 0.0037$. AV+ cell number: 4 h, $P = 0.0184$;

8 h, $P = 0.0261$; 12 h, $P = 0.0001$; 16 h, $P = 0.0001$; 20 h, $P = 0.0001$; 24 h, $P = 0.0001$. PI+ cell number: 8 h, $P = 0.0046$; 12 h, $P = 0.0099$; 24 h, $P = 0.0012$). **e** The number of intact neutrophils remaining. All data are represented as mean ± SD. $n = 3$ independent repeats, *$P < 0.05$, **$P < 0.01$ (Remaining cell number: 12 h, $P = 0.0119$; 16 h, $P = 0.024$; 20 h, $P = 0.0001$; 24 h, $P = 0.0001$. Disappeared cell number: 12 h, $P = 0.0132$; 16 h, $P = 0.0207$; 20 h, $P = 0.0247$). **f** The percentage and number of healthy neutrophils. All data are represented as mean ± SD. $n = 4$ independent repeats, *$P < 0.05$, **$P < 0.01$, ns non-significant (Percentage of healthy cells: 8 h, $P = 0.0499$; 12 h, $P = 0.0115$; 16 h, $P = 0.0001$; 20 h, $P = 0.0005$; 24 h, $P = 0.0008$). **g** The absolute number of healthy neutrophils and neutrophils undergoing apoptosis or lytic cell death. All data are represented as mean ± SD ($n = 4$ independent repeats). **h** LDH release from cell culture supernatants. Data are mean ± SD. $n = 3$ independent repeats, *$P < 0.05$, **$P < 0.01$ (4 h, $P = 0.042$; 8 h, $P = 0.0001$; 12 h, $P = 0.0003$; 16 h, $P = 0.0023$). Statistical significance was examined by unpaired two-sided Student's $t$ test (**d**, **e**, **f**, **h**). Source data are provided as a Source Data file.

polymerase involved in DNA repair and one of the main caspase-3 targets in vivo, was also increased in KO neutrophils compared with WT neutrophils (Fig. 2f).

Next, to confirm that neutrophils isolated from inflamed sites behave similarly to BM neutrophils, peritoneal neutrophils isolated

from thioglycolate (TG)-challenged WT and *Gsdme* KO mice were cultured and their patterns of cell death observed and measured as above. Similar to BM neutrophils, neutrophils from the inflamed peritoneal cavities of *Gsdme* KO mice showed an obvious preference to undergo apoptosis compared with those from WT mice (Fig. S5a–f).

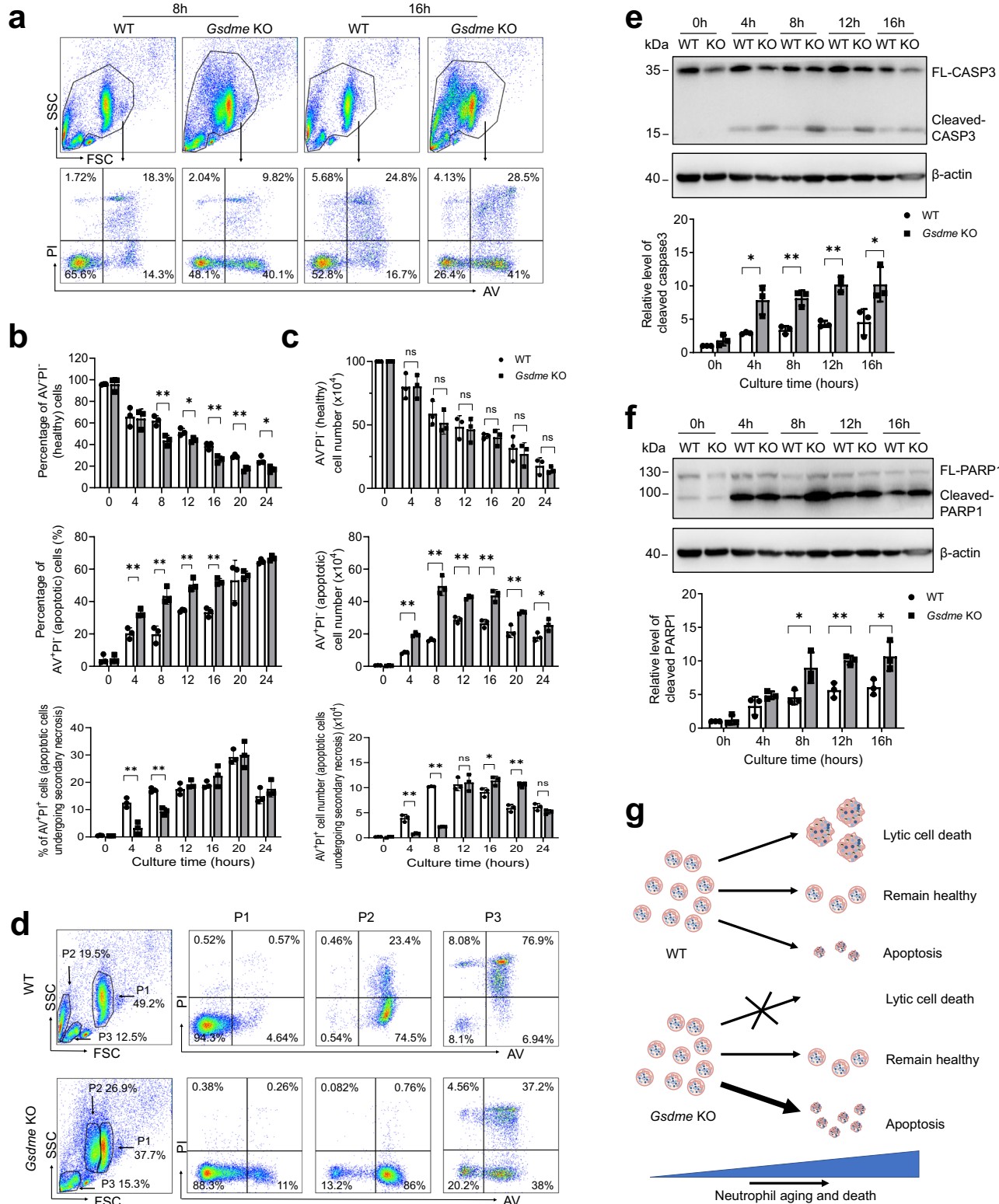

Neutrophils from *Gsdme* KO mice did not swell, again confirming that GSDME deletion abolished lytic cell death in neutrophils (Fig. S5b, c).

To simulate neutrophil death during inflammation under pathophysiological conditions, we analyzed neutrophil death in the presence of various proinflammatory cytokines (Fig. S6). Neither IL-1β nor IL-6 impacted neutrophil death. Consistent with previous reports[35,36], treatment with TNF-a prompted apoptosis, resulting in a notable reduction in both the total number of intact neutrophils (Fig. S6c) and the count of healthy neutrophils (Fig. S6d, f) after 20 h of culture.

Under all these conditions, *Gsdme* KO neutrophils displayed a similar number of healthy cells to that of the WT neutrophils; however, there was a noticeable shift towards apoptosis in the death of the former (Fig. S6d, f).

Collectively, our results demonstrate that disruption of GSDME skews neutrophil death to apoptosis, potentially acting as a master switch controlling the mode of neutrophil death (Fig. 2g). A previous study demonstrated that GSDME, but not GSDMD, drives neutrophil lysis following activation of extrinsic and intrinsic apoptosis[37]. Our

**Fig. 2 | GSDME disruption skewed programmed neutrophil death to apoptosis.**
**a** Representative flow cytometry plots of neutrophils undergoing spontaneous cell death. Results are representative of at least three biological replicates. **b** The percentage of AV⁻PI⁻, AV⁺PI⁻, and AV⁺PI⁺ cells at the indicated time points. All data are represented as mean ± SD. $n = 3$ independent repeats, *$P < 0.05$, **$P < 0.01$, ns non-significant by unpaired two-sided Student's $t$ test (Percentage of AV⁻PI⁻cells: 8 h, $P = 0.0082$; 12 h, $P = 0.0279$; 16 h, $P = 0.0083$; 20 h, $P = 0.001$; 24 h, $P = 0.0288$. Percentage of AV⁺PI⁻ cells: 4 h, $P = 0.0064$; 8 h, $P = 0.0049$; 12 h, $P = 0.0022$; 16 h, $P = 0.0007$. Percentage of AV⁺PI⁺ cells: 4 h, $P = 0.0048$; 8 h, $P = 0.001$). **c** The number of AV⁻PI⁻, AV⁺PI⁻, and AV⁺PI⁺ cells at the indicated time points. All data are represented as mean ± SD. $n = 3$ independent repeats, *$P < 0.05$, **$P < 0.01$, ns non-significant by unpaired two-sided Student's $t$ test (Number of AV⁺PI⁻ cells: 4 h, $P = 0.0001$; 8 h, $P = 0.0006$; 12 h, $P = 0.0004$; 16 h, $P = 0.0024$; 20 h, $P = 0.0059$; 24 h, $P = 0.0455$. Number of AV⁺PI⁺ cells: 4 h, $P = 0.0017$; 8 h, $P = 0.0001$; 16 h, $P = 0.0335$; 20 h, $P = 0.0008$). **d** Representative scatter plots and AV/PI staining of different populations on scatter plots of WT and *Gsdme* KO neutrophils cultured for 8 h. Results are representative of at least three biological replicates. **e** Caspase-3 cleavage during neutrophil spontaneous death. Western blot results are representative of at least three independent experiments. Relative amounts of cleaved caspase-3 were quantified based on densitometry analysis using NIH ImageJ software. Results are the means (±SD) of three independent experiments. *$P < 0.05$, **$P < 0.01$ versus WT neutrophils by two-sided Student's -test (4 h, $P = 0.0168$; 8 h, $P = 0.0032$; 12 h, $P = 0.0010$; 16 h, $P = 0.039$). **f** Cleavage of PARP1 during neutrophil spontaneous death. Western blot results are representative of at least three independent experiments. Relative amounts of cleaved PARP1 were quantified based on densitometry. Results are the means (±SD) of three independent experiments. *$P < 0.05$, **$P < 0.01$ versus WT neutrophils by two-sided Student's $t$-test (8 h, $P = 0.386$; 12 h, $P = 0.004$; 16 h, $P = 0.0319$). **g** Summary of the mode of programmed spontaneous death of WT and *Gsdme* KO neutrophils. Source data are provided as a Source Data file.

observation of decreased lytic cell death in GSDME-deficient neutrophils aligns nicely with these results. However, our current study primarily investigates the role of GSDMD/E in spontaneous neutrophil death and the resolution of inflammation - a domain previously unexplored.

## GSDMD and GSDME have different functions in neutrophil spontaneous death

GSDMD is also implicated in neutrophil death[14]. In neutrophils, GSDMD can be cleaved by caspase-11[38], neutrophil elastase (ELANE)[14,39], and cathepsin G[40]. Consistently, GSDMD cleavage was detected during spontaneous neutrophil death (Fig. S7a). We next assessed patterns of death of WT and *Gsdmd* KO neutrophils through AV/PI staining and morphological analysis as described in Fig. 1b. The absolute number of neutrophils was counted by FACS analysis using counting beads (Fig. S7b). Consistent with our previous study[14], spontaneous death was delayed in GSDMD-deficient neutrophils. After 16 h of culture, 30% of GSDMD-deficient neutrophils remained healthy, compared with only ~20% of WT neutrophils (Fig. S7c). However, GSDMD disruption did not affect the mode of neutrophil death, with the ratio of apoptotic to lytic cells remaining unaltered in GSDMD-deficient neutrophils at each time points during spontaneous death (Fig. S7d, e). The percentage of neutrophils undergoing lytic cell death peaked at about 16 h of culture, with 30–40% lytic cells detected in both WT and *Gsdmd* KO neutrophils (the ratio of apoptotic to lytic cells ≈1.8) (Fig. S7e). By contrast, almost no *Gsdme* KO neutrophils were lytic (Figs. 1–2). Collectively, these results suggest that during neutrophil spontaneous death, GSDME deficiency - but not GSDMD deficiency - abolishes lytic cell death and skews cell death from lytic to non-lytic or apoptotic without changing overall numbers of healthy cells (Fig. 2g). By contrast, GSDMD disruption increases healthy neutrophil numbers but does not alter the mode of neutrophil death (Fig. S7f).

## GSDME is cleaved during neutrophil death via a PR3/caspase-3 axis

Previous studies showed that GSDME is activated by apoptotic caspase-3, driving tumor cell pyroptosis and anti-tumor immunity[28,41,42]. Chen et al. demonstrated that both extrinsic and intrinsic apoptosis pathways activate caspase-3 and GSDME in neutrophils. Upon infection with Yersinia, RIPK1 promotes caspase-3–dependent GSDME activation, leading to neutrophil pyroptosis[37]. Next, we investigated the mechanism of GSDME cleavage during spontaneous neutrophil death. A specific caspase-3 inhibitor (Z-DEVE-FMK) and pan-caspase inhibitor (QVD-OPH), but not caspase 1, 8, or 9 inhibitors, significantly suppressed GSDME cleavage (Fig. 3a), suggesting that caspase-3 cleaved GSDME in aging neutrophils. We previously reported that neutrophil serine protease PR3, which is released from granules to the cytosol during neutrophil aging, cleaves caspase 3 to modulate neutrophil death[43]. Thus, PR3 may cleave and activate caspase 3, which in turn processes GSDME in aging neutrophils. To test this, we treated neutrophils with the serine protease inhibitor DFP and, as predicted, DFP treatment significantly reduced GSDME cleavage (Fig. 3b). Additionally, caspase-3 cleaved mouse recombinant GSDME overexpressed in HEK293T cells (Fig. 3c). PR3, but not ELANE or cathepsin G, via cleavage and activation of endogenously expressed caspase-3 in HEK293T cells, could also cleave GSDME to produce active NT-GSDME. The cleavage of GSDME was also observed in aging human neutrophils. Similarly, this process could be inhibited by both serine protease inhibitor DFP and caspase-3 inhibitor Z-DEVE-FMK (Fig. S8). Taken together, GSDME cleavage in aging neutrophils is at least partially mediated by a PR3/caspase-3 axis (Fig. 3d). Additionally, we measured GSDME cleavage in neutrophils isolated from HIEC-challenged mice and verified that GSDME cleavage indeed occurred in vivo. Of note, under inflammatory conditions in vivo, dead neutrophils and extracellular GSDME were rapidly cleared. Consequently, the amount of cleaved GSDME in intact neutrophils collected from the inflamed site was expectedly lower than that detected in cultured neutrophils (Fig. 3e).

## GSDME deletion does not alter healthy neutrophil numbers at sites of inflammation

In in vitro neutrophil death assays, GSDME dictated the pattern of neutrophil death without affecting healthy cell numbers (Figs. 1, 2). We next investigated how GSDME regulates neutrophil death in vivo. First, purified BM neutrophils from WT (CD45.1) and *Gsdme* KO (CD45.2; CD45.2 WT as a control) mice were mixed (1:1) and adoptively transferred into septic GFP-expressing (*B6 ACTb-EGFP*) recipient WT mice (Fig. S9a). After 15 h of adoptive transfer, neutrophils were collected from the peritoneal cavity and the CD45.2⁺ to CD45.1⁺ neutrophil ratio was determined in the GFP-negative donor neutrophil population. The ratio remained at 1 when either *Gsdme* KO or WT CD45.2 neutrophils were mixed and transferred with WT CD45.1 neutrophils (Fig. S9b, c). In a similar experiment, to minimize possible differences caused by the genetic background (CD45.1 vs CD45.2 mice), we used WT and KO mice from the same background (CD45.2). Also, to check whether neutrophils isolated from inflammatory sites behave similarly to BM neutrophils, we isolated neutrophils from the inflamed peritoneal cavities of TG-challenged hosts. WT and KO neutrophils were then labelled with either CFSE or Snarf-1, mixed 1:1, and adoptively transferred into septic WT hosts (Fig. S9d). 15 h after transfer, neutrophils were collected from the peritoneal cavity and the WT to KO neutrophil ratio calculated by FACS. As before, the ratio did not alter even after 15 h (Fig. S9e, f). At sites of inflammation, dying neutrophils are quickly removed by macrophages and monocytes, and most remaining neutrophils are intact healthy neutrophils. Thus, the constant ratio of adoptively transferred WT and KO neutrophils confirmed that GSDME deletion does not alter healthy neutrophil numbers in vivo under septic conditions.

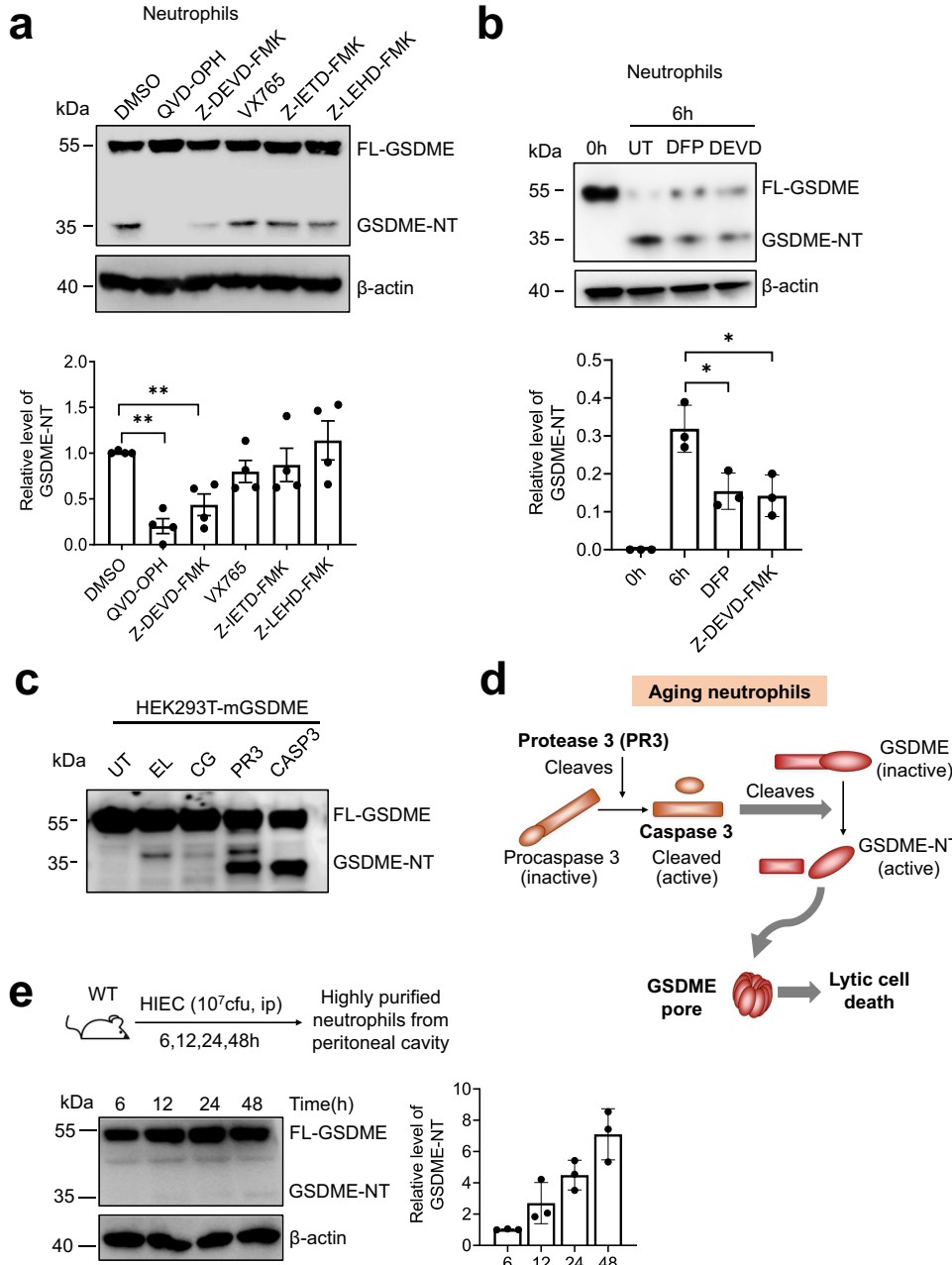

**Fig. 3 | GSDME cleavage during programed spontaneous neutrophil death was mediated by PR3 and caspase-3. a** GSDME cleavage in neutrophils treated with caspase inhibitors. Neutrophils from WT mouse bone marrow were treated with pan-caspase (QVD-OPH 100 μM), caspase-3 (Z-DEVD-FMK 100 μM), caspase-1 (VX765 25 μM), caspase-8 (Z-IETD-FMK 10 μM), or caspase-9 (Z-LEHD-FMK 20 μM) inhibitors for 16 h. β-actin was used as a protein loading control. Results are representative of three biological replicates. Relative levels of cleaved GSDME-NT were quantified based on densitometry. Results are means (±SD). n = 4 independent repeats. **P < 0.01 versus DMSO-treated neutrophils by two-sided Student's t-test (QVD-OPH, P = 0.0001; Z-DEVD-FMK, P = 0.0029). **b** GSDME cleavage in neutrophils treated with serine protease inhibitor diisopropyl fluorophosphate (DFP). Neutrophils were treated with serine protease DFP (100 μM) or Z-DEVD-FMK (as a positive control) for 6 h. Representative immunoblots of three independent experiments are shown. The densitometry results are the means (±SD) of three independent experiments. *P < 0.05 versus DMSO-treated neutrophils by two-sided Student's t-test (DFP, P = 0.022; Z-DEVD-FMK, P = 0.0211). **c** Cleavage of GSDME by

serine proteases and caspase-3. Mouse GSDME was overexpressed in HEK293T cells. Cell lysates were incubated with the indicated proteases or caspase-3 for 1 h. GSDME cleavage was assessed by immunoblotting. UT untreated, EL elastase, CG cathepsin G, PR3 proteinase 3. Results are representative of three independent experiments. **d** Summary of GSDME cleavage during programed spontaneous neutrophil death. In aging neutrophils, PR3 is released from the granules, leading to cleavage and activation of procaspase-3. Active caspase-3 in turn cleaves GSDME to generate GSDME-NT, which targets the plasma membrane to induce membrane pore formation and lytic cell death. **e** GSDME cleavage in neutrophils during inflammation. Mice were challenged with heat-inactivated *E. coli*. (HIEC, $10^7$ cfu) for indicated time periods. GSDME cleavage in purified neutrophils was assessed by western blotting as described above. Representative immunoblots of three independent experiments are shown. The densitometry results are the means (±SD) of three independent experiments. Source data are provided as a Source Data file.

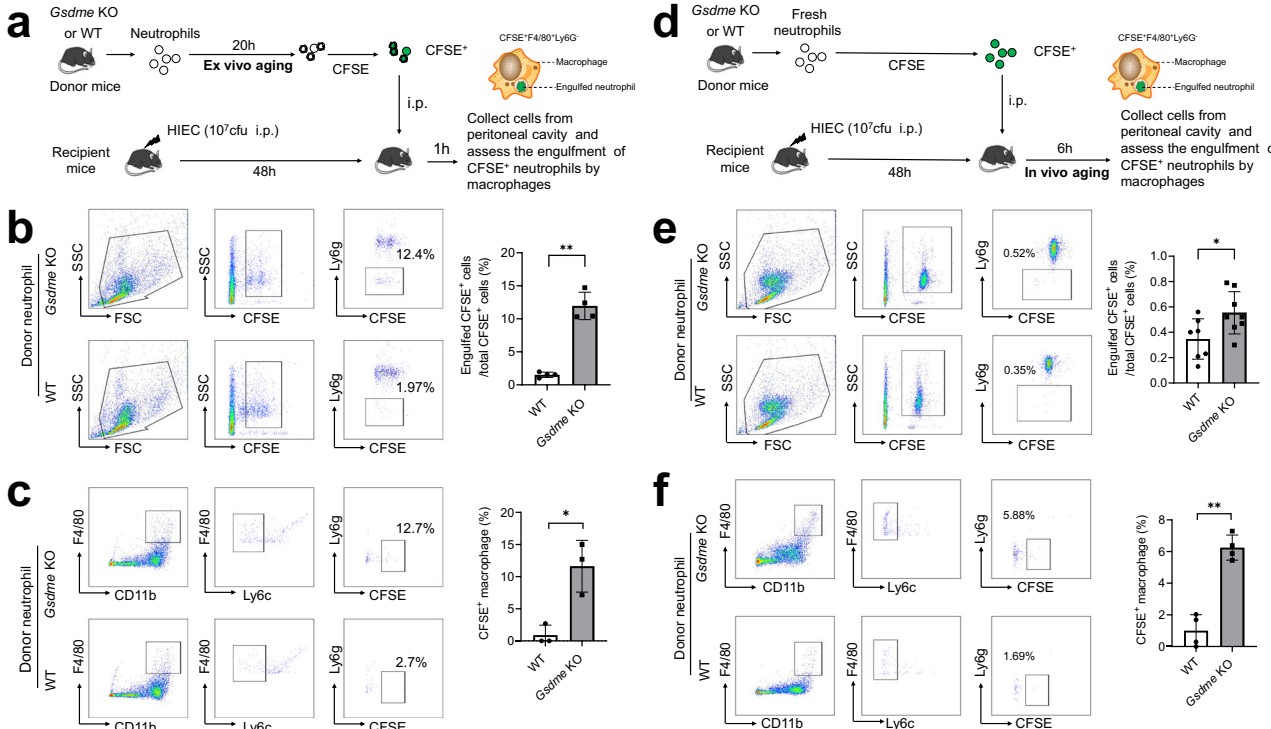

**Fig. 4 | Genetic ablation of neutrophil GSDME enhanced efferocytosis in peritonitis. a** Experimental scheme for assessing efferocytosis of ex vivo-aged neutrophils in peritonitis. Neutrophils from WT and *Gsdme* KO mice were cultured for 20 h and stained with 1 μM CFSE for 10 min. CFSE-labeled aged neutrophils were intraperitoneally injected into recipient mice challenged with heat-inactivated *E. coli*. (HIEC, 10$^7$ cfu) for 48 h. 1 h post adoptive transfer, the peritoneal cavity was lavaged to collect cells to assess engulfment of CFSE$^+$ neutrophils by macrophages. **b** The percentage of adoptively transferred neutrophils engulfed by macrophages was assessed by flow cytometry. After being engulfed, adoptively transferred neutrophils became ly6g negative but remained CFSE positive. All data are represented as mean ± SD, *n* = 4 mice, **P* < 0.01 (*P* = 0.0001). **c** The percentage of macrophages engulfing at least one apoptotic neutrophil was assessed by flow cytometry. In the peritoneal cavity, F4/80$^+$CD11b$^+$Ly6c$^-$ macrophages could engulf

apoptotic neutrophils. After engulfing adoptively transferred neutrophils, macrophages became CFSE positive. All data are represented as mean ± SD, *n* = 3 mice, **P* < 0.05 (*P* = 0.0126). **d** Experimental scheme for assessing efferocytosis of in vivo-aged neutrophils in peritonitis. Fresh neutrophils from WT and *Gsdme* KO mice were stained with 1 μM CFSE for 10 min. CFSE-labeled fresh neutrophils were intraperitoneally injected into recipient mice challenged with HIEC for 48 h. 6 h post adoptive transfer, the peritoneal cavity was lavaged to collect cells to assess engulfment of CFSE$^+$ neutrophils by macrophages. **e** The percentage of adoptively transferred neutrophils engulfed by macrophages was assessed by flow cytometry. All data are represented as mean ± SD, *n* = 7−8 mice, **P* < 0.05 (*P* = 0.0293). **f** The percentage of macrophages engulfing at least one apoptotic neutrophil was assessed by flow cytometry. All data are represented as mean ± SD, *n* = 4 mice; **P* < 0.01 (*P* = 0.0002). Source data are provided as a Source Data file.

## GSDME disruption increases neutrophil efferocytosis by macrophages

It is well documented that AV (externalized phosphatidylserine, PS)-positive apoptotic neutrophils are quickly cleared by macrophages via efferocytosis via the PS receptor (PSR)[44–48]. We next investigated whether the observed increase in apoptosis would augment efferocytosis in *Gsdme* KO mice. We conducted an adoptive transfer experiment in which fluorescently-labelled aged neutrophils from WT and KO mice were transferred into the peritoneal cavities of septic WT hosts (Fig. 4a). In this setup, neutrophils were cultured for 20 h to allow them to age and undergo ex vivo cell death (ex vivo aging). Efferocytosis was quantified 1 h after neutrophil transfer, as efferocytosis occurs soon after neutrophils undergo apoptosis[23]. Macrophages that engulfed labeled neutrophils would gain CFSE staining from neutrophils, allowing quantification by flow cytometry. Significantly more *Gsdme* KO neutrophils were efferocytosed than WT neutrophils (Fig. 4b). The percentage of CFSE$^+$ cells engulfed by macrophages was over 10-times higher when *Gsdme* KO neutrophils were transferred compared with WT (Fig. 4b). Similarly, more macrophages were CFSE$^+$ after *Gsdme* KO neutrophil transfer compared with after WT neutrophil transfer (Fig. 4c).

In another setup, fresh neutrophils from WT and *Gsdme* KO mice were CFSE labelled and adoptively transferred to septic WT recipient hosts. Efferocytosis was quantified six hours after transfer (Fig. 4d).

Here, the transferred fresh neutrophils aged and underwent in vivo cell death at the site of inflammation (in vivo aging). Again, efferocytosis was much greater when *Gsdme* KO neutrophils were transferred compared with after WT transfer (Fig. 4e, f). Collectively, these results confirm that GSDME-deficient neutrophils undergo more apoptosis and that apoptotic neutrophils are quickly cleared by macrophages via efferocytosis.

## GSDME deficiency triggers anti-inflammatory responses

GSDME deficiency skewed neutrophil death to apoptosis, which in turn augmented efferocytosis. Apoptosis can elicit anti-inflammatory responses via efferocytosis[18–20]. Thus, we investigated whether GSDME disruption attenuated inflammatory responses during infection. To this end, we induced peritonitis in WT and *Gsdme* KO mice using heat-inactivated *E. coli* (HIEC, instead of live *E.coli* to eliminate any effect associated with host bactericidal capability) and investigated the inflammatory milieu (Fig. 5a). As expected, IL-10, an important anti-inflammatory cytokine released during efferocytosis, was upregulated in *Gsdme* KO mice (Fig. 5b). By contrast, production of IL-1β, a pro-inflammatory cytokine, was reduced in *Gsdme* KO mice compared with WT (Fig. 5b), suggesting an attenuated inflammatory response in KO mice. Consistently, the percentage and absolute number of neutrophils accumulated in the peritoneal cavity were less in *Gsdme* KO mice than WT counterparts (Fig. 5c). Of note, the

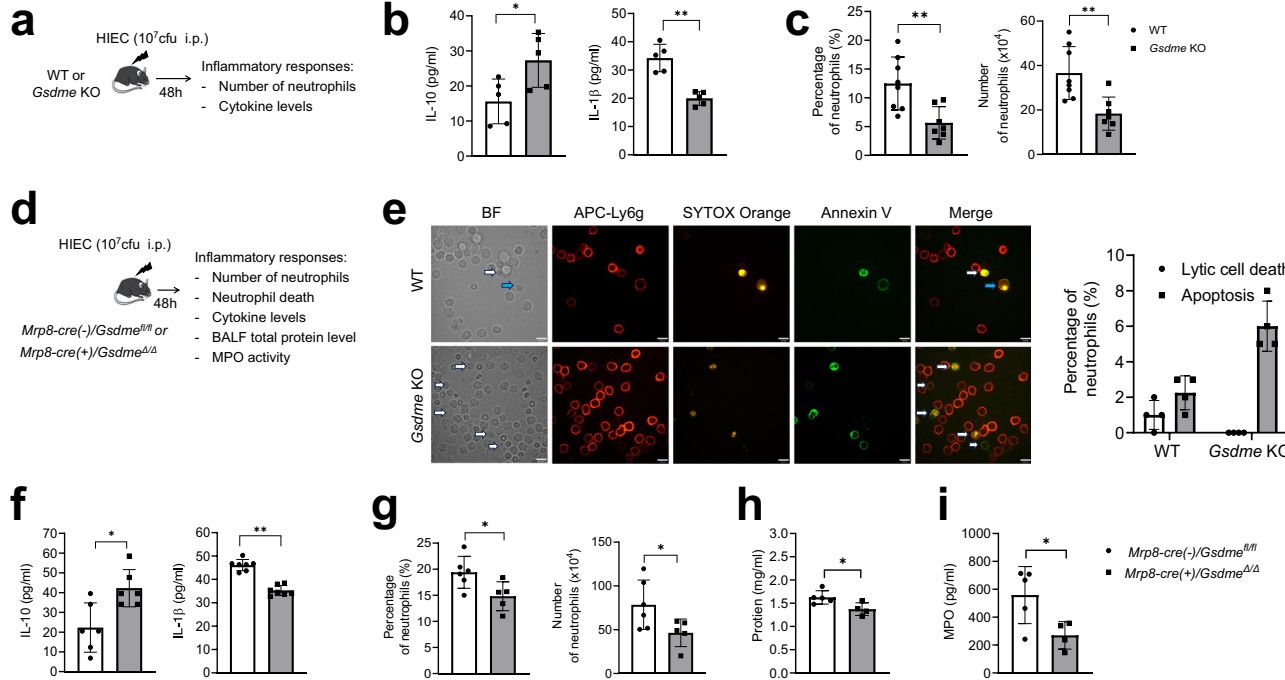

**Fig. 5 | Genetic ablation of neutrophil GSDME suppressed inflammatory responses in peritonitis. a** Schematic showing the experimental design for assessing inflammatory responses of WT and *Gsdme* whole-body KO mice during peritonitis. **b** IL-10 and IL-1β levels in the peritoneal lavage fluid. All data are represented as mean ± SD, *n* = 5 mice, *P < 0.05, **P < 0.01 (IL-10, P = 0.030; IL-1β, P = 0.0004). **c** Neutrophil percentage and number in the peritoneal cavity. All data are represented as mean ± SD, *n* = 7–8 mice, **P < 0.01 (Percentage of neutrophils, P = 0.0048; Number of neutrophils, P = 0.004). **d** Schematic showing the experimental design for assessing inflammatory responses of WT (*Mrp8-cre(-)/Gsdme^fl/fl*) and neutrophil-specific conditional *Gsdme* KO (*Mrp8-cre(+)/Gsdme^Δ/Δ*) mice during peritonitis. **e** In vivo neutrophil death. Lavage fluid from the peritoneal cavity was directly stained with APC-Ly6g to label neutrophils, and with Annexin V (AV) and SYTOX Orange to mark dead cells, all without centrifugation. The cells were subsequently fixed and immediately photographed. The images

are representative of at least three independent experiments. Blue arrowheads indicate lytic dead neutrophils and white arrowheads indicate apoptotic neutrophils. The scale bar represents 10 μm. Data are presented as the mean ± SD, *n* = 3–4 independent repeats. **f** IL-10 and IL-1β levels in the peritoneal lavage fluid. All data are represented as mean ± SD, *n* = 6 mice, *P < 0.05, **P < 0.01 (IL-10, P = 0.0108; IL-1β, P = 0.0001). **g** Percentage and number of neutrophils in the peritoneal cavity. All data are represented as mean ± SD, *n* = 5–6 mice, *P < 0.05 (Percentage of neutrophils, P = 0.0299; Number of neutrophils, P = 0.0452). **h** Total protein level in the peritoneal lavage fluid. All data are represented as mean ± SD, *n* = 4–5 mice, *P < 0.05 (P = 0.0308). **i** MPO level in the peritoneal lavage fluid. All data are represented as mean ± SD, *n* = 4–5 mice, *P < 0.05 (P = 0.0368). Statistical significance was examined by unpaired two-sided Student's *t* test (b, c, e, f, h, I, l, m, n, o). Source data are provided as a Source Data file.

percentage and numbers of neutrophils, monocytes, B cells, and T cells in the BM were similar between WT and *Gsdme* KO mice (Fig. S10a). Additionally, GSDME deficiency did not alter macrophage, monocyte, B cell, or T cell numbers in the peritoneal cavity (Fig. S10b). Finally, we checked if the reduced neutrophil accumulation in *Gsdme* KO mice was due to a defect in recruitment. When the number and percentage of neutrophils in the peritoneal cavity were analyzed at an early time point (6 h post HIEC challenge) when neutrophil death had not yet started, there was no difference between WT and KO mice (Fig. S10c). These observations confirm that the reduction in neutrophils in the peritoneal cavity of *Gsdme* KO mice 48 h post HIEC challenge was due to the overall attenuated host inflammatory response triggered by enhanced efferocytosis.

The attenuated inflammatory response observed in *Gsdme* whole-body KO mice might be caused by disruption of GSDME in other cell types. To exclude this possibility and to confirm that the phenotype was mediated by neutrophil GSDME, we generated a neutrophil-specific conditional *Gsdme* KO mouse (*Mrp8-cre(+)/Gsdme^Δ/Δ* or cKO) (Fig. S11a–c). Similar to whole-body KO mice, the total leukocyte and differential blood counts were unchanged after ablation of GSDME from neutrophils (Fig. S11d). *Mrp8-cre(-)/Gsdme^fl/fl* (WT controls) and *Mrp8-cre(+)/Gsdme^Δ/Δ* littermates were challenged with HIEC to induce peritonitis (Fig. 5d). First, we examined the nature of neutrophil death in vivo and confirmed that GSDME disruption elicited an increase in neutrophil apoptosis under physiological inflammatory conditions (Fig. 5e). We then measured the inflammatory responses of the host.

Similar to whole-body KO mice, *Mrp8-cre(+)/Gsdme^Δ/Δ* mice produced more anti-inflammatory IL-10 and less pro-inflammatory IL-1β than their WT littermates (Fig. 5f). Neutrophil accumulation in the peritoneal cavity was also attenuated in *Mrp8-cre(+)/Gsdme^Δ/Δ* mice compared with WT littermates (Fig. 5g), while the number of peritoneal macrophages, monocytes, T cells, and B cells were the same in the two groups (Fig. S12a). Similarly, the percentage and number of BM neutrophils, monocytes, B cells, and T cells were also similar between HIEC-challenged WT and cKO mice (Fig. S12b). As observed in whole-body *Gsdme* KO mice, peritoneal neutrophil numbers at early time points were unaffected by GSDME deletion in neutrophils (Fig. S12c), indicating unaffected neutrophil recruitment. Suppression of HIEC-induced inflammation in the cKO mice was also evidenced by a reduction in the total BALF protein level (Fig. 5h), an indicator of vascular leakage, as well as by a decreased BALF level of myeloperoxidase (MPO) (Fig. 5i), a component of neutrophil granules and a crucial parameter of pulmonary inflammation. Together, these results demonstrate that neutrophil-specific GSDME deletion suppresses host inflammatory responses during infection and inflammation.

## GSDME dictates the host inflammatory response in LPS-induced acute lung injury

Next, we studied the role of GSDME in regulating host inflammatory responses in a murine acute lung injury model. We challenged WT and *Gsdme* KO mice with intra-tracheal instillation of *E. coli*-derived LPS and observed the inflammatory milieu in the lungs after 24 h (Fig. 6a).

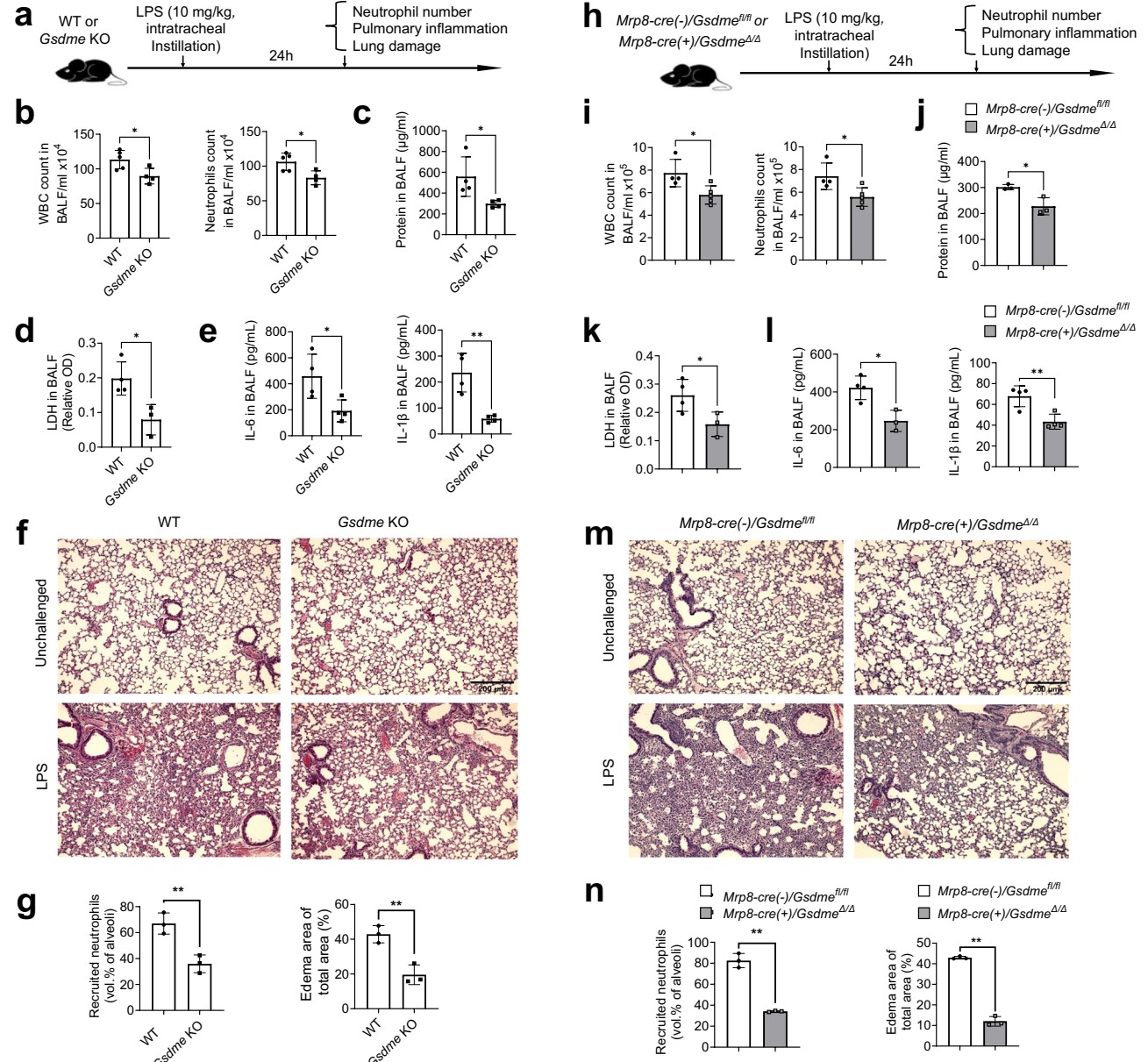

**Fig. 6 | GSDME disruption in neutrophils attenuated inflammation and alleviated lung injury during LPS-induced pneumonia. a** Schematic showing the experimental design for LPS-induced acute lung injury. **b** Quantification of total leukocytes and neutrophils in the BALF. All data are represented as mean ± SD, $n = 4–5$ mice/group, *$P < 0.05$ (WBC, $P = 0.0247$; Neutrophil, $P = 0.0194$). **c** Total protein in the BALF. Data are represented as mean ± SD, $n = 4$ mice/group, *$P < 0.05$ ($P = 0.0350$). **d** LDH release in the BALF. Data are represented as mean ± SD, $n = 3–4$/group, *$P < 0.05$ ($P = 0.0201$). **e** IL-6 and IL-1β levels in BALF. Data are represented as mean ± SD, $n = 4$ mice/group, *$P < 0.05$, **$P < 0.01$ (IL-6, $P = 0.0309$; IL-1β, $P = 0.0033$). **f** Hematoxylin and eosin (H&E) staining of lung sections. Results are representative of three biological replicates. **g** Lung inflammation and damage assessed as neutrophil recruitment (vol % of alveoli) and edema area of lung. Data are represented as mean ± SD, $n = 4–5$ mice/group, **$P < 0.01$ (Recruited neutrophils, $P = 0.0072$; Edema area of total area, $P = 0.0058$). **h** Experimental scheme for assessing LPS-induced pulmonary inflammation and associated lung injury in WT and neutrophil-specific *Gsdme* conditional KO mice. **i** Quantification of total leukocytes and neutrophils in the BALF. Data are represented as mean ± SD, $n = 4–5$ mice/group, *$P < 0.05$ (WBC, $P = 0.0238$; Neutrophil, $P = 0.0276$). **j** Total protein in the BALF. Data are represented as mean ± SD, $n = 3–4$ mice/group, *$P < 0.05$ ($P = 0.0214$). **k** LDH release in the BALF. Data are represented as mean ± SD, $n = 3–4$ mice/group, *$P < 0.05$ ($P = 0.0487$). **l** IL-6 and IL-1β levels in the BALF. Data are represented as mean ± SD, $n = 3–4$ mice/group, *$P < 0.05$, **$P < 0.01$ (IL-6, $P = 0.0123$; IL-1β, $P = 0.0074$). **m** H&E staining of lung sections. Results are representative of three biological replicates. **n** Pulmonary inflammation and tissue damage measured as neutrophil recruitment and edema area of lung. Data are represented as mean ± SD, $n = 3$ mice/group, **$P < 0.01$ ($P = 0.0001$). Statistical significance was examined by unpaired two-sided Student's *t* test (b, c, d, e, g, i, j, k, l, n). Source data are provided as a Source Data file.

Inflammation was attenuated in *Gsdme* KO mice, as depicted by reduced total leukocyte and neutrophil accumulation in the alveolar spaces (Fig. 6b). Total protein levels in bronchoalveolar lavage fluid (BALF), which is an indicator of lung damage, were lower in *Gsdme* KO mice (Fig. 6c). Similarly, levels of LDH were also reduced in KO mice compared with WT counterparts (Fig. 6d). Additionally, pro-inflammatory cytokines IL-6 and IL-1β were reduced in KO mice compared with their WT counterparts (Fig. 6e). Finally, histopathological analysis of infected lungs showed attenuated lung injury and neutrophil accumulation in *Gsdme* KO mice compared with WT (Fig. 6f–g). Similar attenuated inflammatory responses were also observed 48 h after LPS challenge (Fig. S13a–h).

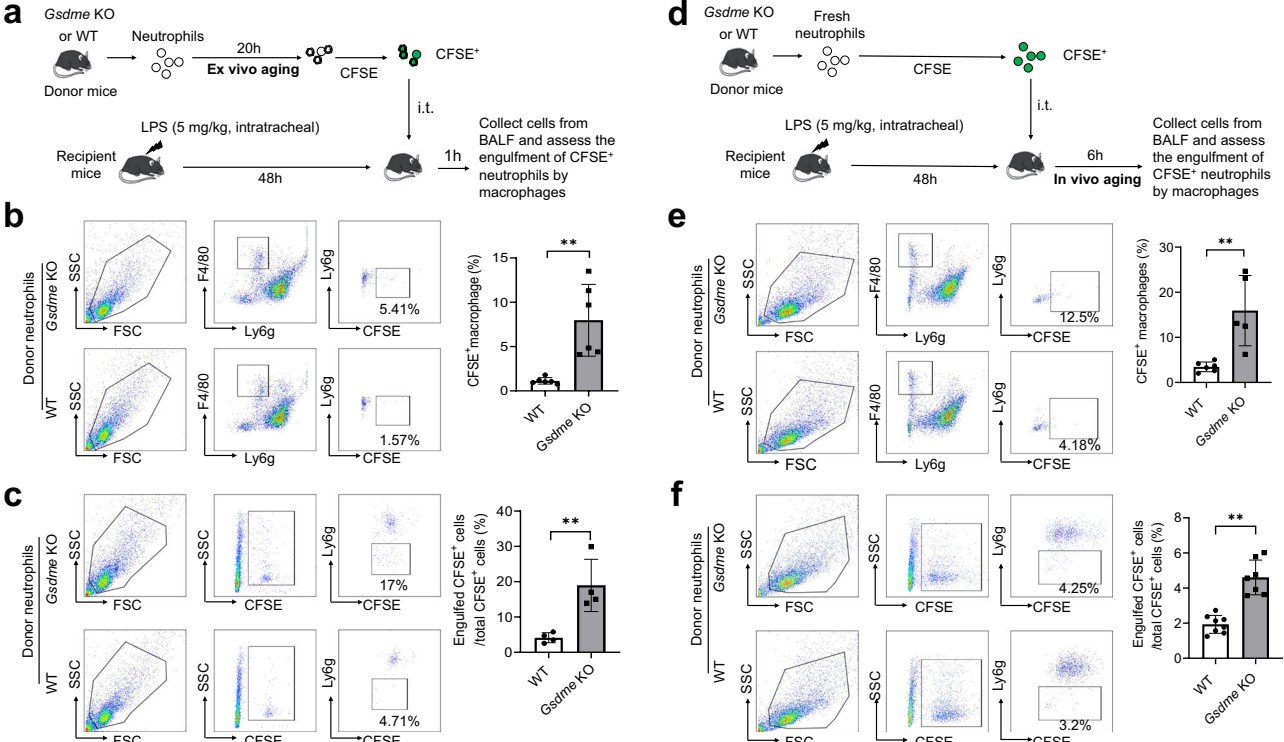

**Fig. 7 | GSDME disruption in neutrophils enhanced efferocytosis in the inflamed lungs. a** Experimental scheme for assessing efferocytosis of ex vivo-aged neutrophils in LPS-induced acute lung injury. BM neutrophils from WT and *Gsdme* KO mice were cultured for 20 h and stained with 1 μM CFSE for 10 min. CFSE-labeled aged neutrophils were intratracheally injected into recipient mice challenged with 5 mg/kg LPS. 1 h post adoptive transfer, lungs were lavaged to collect cells to assess the engulfment of CFSE⁺ neutrophils by macrophages. **b** The percentage of macrophages engulfing at least one apoptotic neutrophil was assessed by flow cytometry as described in Fig. 4. The flow cytometric images are representative of at least three independent experiments. All data are represented as mean ± SD, *n* = 6 mice, **P < 0.01 by two-sided Student's *t*-test (*P* = 0.0021). **c** The percentage of adoptively transferred neutrophils engulfed by macrophages was assessed by flow cytometry as described in Fig. 4. The flow cytometric images are representative of at least three independent experiments. All data are represented as mean ± SD,

*n* = 4 mice, **P < 0.01 by two-sided Student's *t*-test (*P* = 0.0077). **d** Experimental scheme for assessing efferocytosis of in vivo-aged neutrophils in LPS-induced acute lung injury. Fresh neutrophils from WT and *Gsdme* KO mice were stained with CFSE. CFSE-labeled fresh neutrophils were intratracheally injected into recipient mice challenged with LPS. 6 h post adoptive transfer, lungs were lavaged to collect cells to assess engulfment of CFSE⁺ neutrophils by macrophages. **e** The percentage of macrophages engulfing at least one apoptotic neutrophil was assessed by flow cytometry. The flow cytometric pictures are representative of at least three independent experiments. All data are represented as mean ± SD, *n* = 5 mice; **P < 0.01 by two-sided Student's *t*-test (*P* = 0.0035). **f** The percentage of adoptively transferred neutrophils engulfed by macrophages was assessed by flow cytometry. The flow cytometric pictures are representative of at least three independent experiments. All data are represented as mean ± SD, *n* = 7 mice, **P < 0.01 by two-sided Student's *t*-test (*P* = 0.0001). Source data are provided as a Source Data file.

To confirm that the phenotype was mediated by GSDME disruption in neutrophils, we induced LPS pneumonia in *Mrp8-cre(-)/Gsdme^{fl/fl}* (WT) and *Mrp8-cre(+)/Gsdme^{Δ/Δ}* littermates and observed the inflammatory milieu (Fig. 6h). As in whole body KO mice, inflammation was reduced in *Mrp8-cre(+)/Gsdme^{Δ/Δ}* mice, as evidenced by diminished total leukocyte and neutrophil accumulation in the alveolar space (Fig. 6i), reduced total protein levels in BALF (Fig. 6j), suppressed LDH secretion (Fig. 6k), and attenuated production of proinflammatory IL-6 and IL-β (Fig. 6l). Consistently, GSDME deletion in neutrophils reduced lung damage as observed histopathologically (Fig. 6m–n).

Finally, we directly assessed efferocytosis of adoptively transferred WT and *Gsdme* KO neutrophils by alveolar macrophages in inflamed lungs (Fig. 7). LPS pneumonia was induced in WT recipient mice. Forty-eight hours post LPS challenge, CFSE-labeled neutrophils from WT or KO mice were adoptively transferred into WT hosts via intra-tracheal injection. Neutrophils underwent aging and death either ex vivo (Fig. 7a–c) or in vivo (Fig. 7d–f). In both cases, similar to in the peritonitis model, GSDME disruption in donor neutrophils significantly augmented efferocytosis, as indicated by the increased percentage of macrophages engulfing donor neutrophils (Fig. 7b, e) and the percentage of donor neutrophils engulfed by macrophages (Fig. 7c, f). Together, these results demonstrate that deletion of GSDME in neutrophils enhances neutrophil efferocytosis

by macrophages, thereby attenuating inflammatory responses and acute lung injury.

## GSDME disruption in neutrophils alleviates lung damage in both bacterial pneumonia and acid aspiration pneumonitis

LPS-induced ALI is a widely used and accepted model for studying host immunity and inflammation-elicited tissue damage. However, suppressing inflammatory responses may not necessarily be a valid therapeutic strategy in pneumonia patients. Pathogens and pathogen-associated molecular patterns (PAMPs) elicit significant inflammatory responses that not only contribute to pneumonia-elicited lung damage but may also be essential for host defenses against invading bacterial pathogens. Our in vitro and in vivo experiments revealed that GSDME regulates neutrophil death to dictate the host inflammatory response. We next examined the role of GSDME in modulating host immune response in a mouse *Staphylococcus aureus*, a gram-positive bacteria, induced pneumonia model (Fig. 8a). In accordance with the LPS-induced lung injury model, the deletion of GSDME in neutrophils did not affect the initial neutrophil recruitment at 8 h post-infection. However, at 24 and 48 h post-infection, Mrp8-cre(+)/Gsdme^{Δ/Δ} mice exhibited a reduction in total WBCs and neutrophil recruitment compared to their WT littermates (Fig. 8b). Similar to what was observed in the LPS-induced ALI model, neutrophil-specific GSDME disruption also

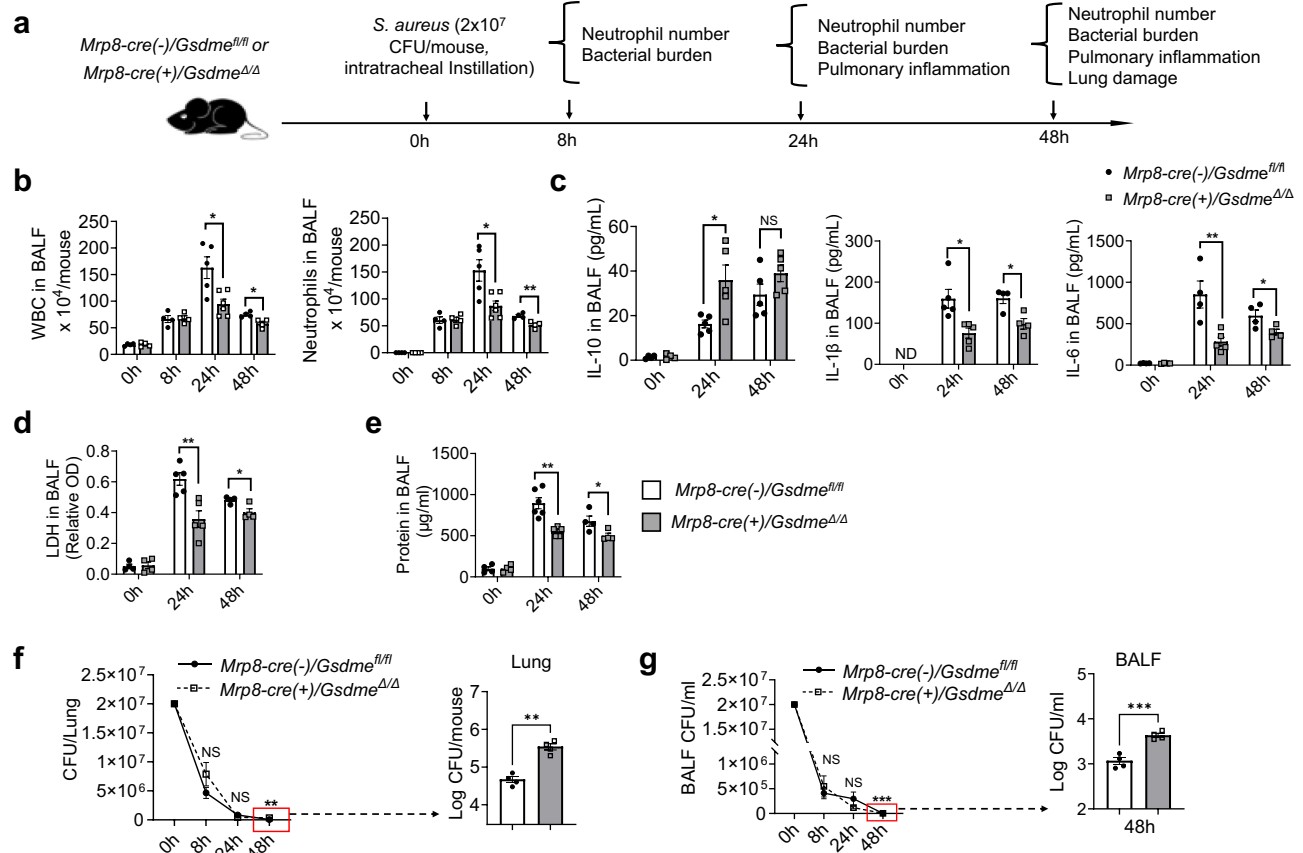

**Fig. 8 | GSDME disruption in neutrophils attenuated inflammation during *S. aureus*-induced pneumonia. a** Experimental scheme for assessing S. aureus-induced pulmonary inflammation and associated lung injury in WT and neutrophil-specific *Gsdme* conditional KO mice. Pneumonia was induced in neutrophil-specific *Gsdme* conditional KO (*Mrp8-cre(+)/Gsdme^Δ/Δ^*) and their WT (*Mrp8-cre(-)/Gsdme^fl/fl^*) littermates using *S. aureus*. Neutrophil recruitment, pulmonary inflammation, and lung damage were evaluated at different time points post-infection. **b** Quantification of total leukocytes and neutrophils in the BALF from *Mrp8-cre(-)/Gsdme^fl/fl^* and *Mrp8-cre(+)/Gsdme^Δ/Δ^* mice post-pneumonia induction at indicated time points. Data are represented as mean ± SEM (*n* = 4–6 mice /group). \**P* < 0.05, \*\**P* < 0.01 by two-sided Student's *t*-test (WBC: 24 h, *P* = 0.0115; 48 h, *P* = 0.0125. Neutrophils: 24 h, *P* = 0.0107; 48 h, *P* = 0.0058). **c** IL-1β, IL-1, and IL-10 levels in the BALF from mice infected with *S. aureus*. Data are shown as mean ± SEM (*n* = 4–6 mice /group). \**P* < 0.05. ND not detectable (IL-10,

*P* = 0.0221. IL-1β: 24 h, *P* = 0.0118; 48 h, *P* = 0.0140. IL-6: 24 h, *P* = 0.0081; 48 h, *P* = 0.0435). **d** LDH release in BALF post-infection. Data are represented as mean ± SEM (*n* = 4–6 mice /group). \**P* < 0.05 by two-sided Student's *t*-test (24 h, *P* = 0.0049; 48 h, *P* = 0.0215). **e** Total protein concentration in the BALF. Data are shown as mean ± SEM (*n* = 4–6 mice/group). \**P* < 0.05 by two-sided Student's *t*-test (24 h, *P* = 0.009; 48 h, *P* = 0.0453). **f** Bacterial burden in the lungs of *Mrp8-cre(-)/Gsdme^fl/fl^* and *Mrp8-cre(+)/Gsdme^Δ/Δ^* mice post *S. aureus* infection at the specified time points. Data are represented as mean ± SEM (*n* = 4–6 mice/group). \*\**P* < 0.01. NS not significant by two-sided Student's *t*-test (*P* = 0.0035). **g** Bacterial count in the BALF from *Mrp8-cre(-)/Gsdme^fl/fl^* and *Mrp8-cre(+)/Gsdme^Δ/Δ^* mice post *S. aureus* infection at the specified time points. Data are presented as mean ± SEM (*n* = 4–6/ mice group). \*\*\**P* < 0.001. NS not significant by two-sided Student's *t*-test (*P* = 0.0007). Source data are provided as a Source Data file.

led to elevated production of anti-inflammatory cytokine IL-10 and attenuated production of proinflammatory IL-6 and IL-β (Fig. 8c), suppressed LDH secretion (Fig. 8d), as well as reduced total protein levels in BALF (Fig. 8e). Importantly, despite the reduction in neutrophil numbers, GSDME disruption in neutrophils did not compromise the host's overall killing capability; over 99% of bacteria were cleared in both the WT and neutrophil-specific cKO mice (Fig. 8f–g). Intriguingly, in the later stages of bacterial infection, the cKO mice displayed higher CFU in the BALF and the whole lung, indicating that the complete eradication of bacteria was delayed in these mice (Fig. 8f–g). However, this very low dose of bacteria evidently did not elicit any adverse effect in the host (Fig. 8b–e). Consistently, due to alleviated inflammation, GSDME deletion in neutrophils reduced lung damage as observed histopathologically (Fig. 9a, b), and ultimately improved survival of infected mice (Fig. 9c). These results not only provide insight into how neutrophil GSDME modulates their antibacterial function, but also demonstrate that the GSDME-mediated immune modulation is a general mechanism that extends beyond LPS-elicited inflammation.

Finally, we evaluated the therapeutic potential of GSDME inhibition in a sterile, clinically relevant acid aspiration-induced lung injury model (Fig. S14a). Aspiration of the acidic gastric contents significantly risks acute lung injury and life-threatening acute respiratory distress syndrome (ARDS), as it causes chemical injury to the alveolar epithelium and capillary endothelium[49,50] of the tracheobronchial tree and pulmonary parenchyma, eliciting an intense inflammatory response[51], primarily mediated by neutrophils recruited to the lung[52,53]. Indeed, aspiration pneumonitis (Mendelson's syndrome) is a major complication of general anesthesia, accounting for 10–30% of all anesthesia-associated deaths[51,54] and occurring in 1:3000 anesthetic cases, and ~10% of patients hospitalized after a drug overdose develop aspiration pneumonitis. It is also a common occurrence in critically ill, elderly, and unconscious patients[51,55,56]. Acid aspiration can be induced in mice by instilling with hydrochloric acid, leading to acute lung injury that mimics human aspiration pneumonitis[55,57]. Consistent with previous reports[55,58], acid instillation recruited significant numbers of neutrophils to the damaged lungs and acute sterile inflammation (Fig. S14b–f). However, inflammation was attenuated in *Gsdme* KO mice

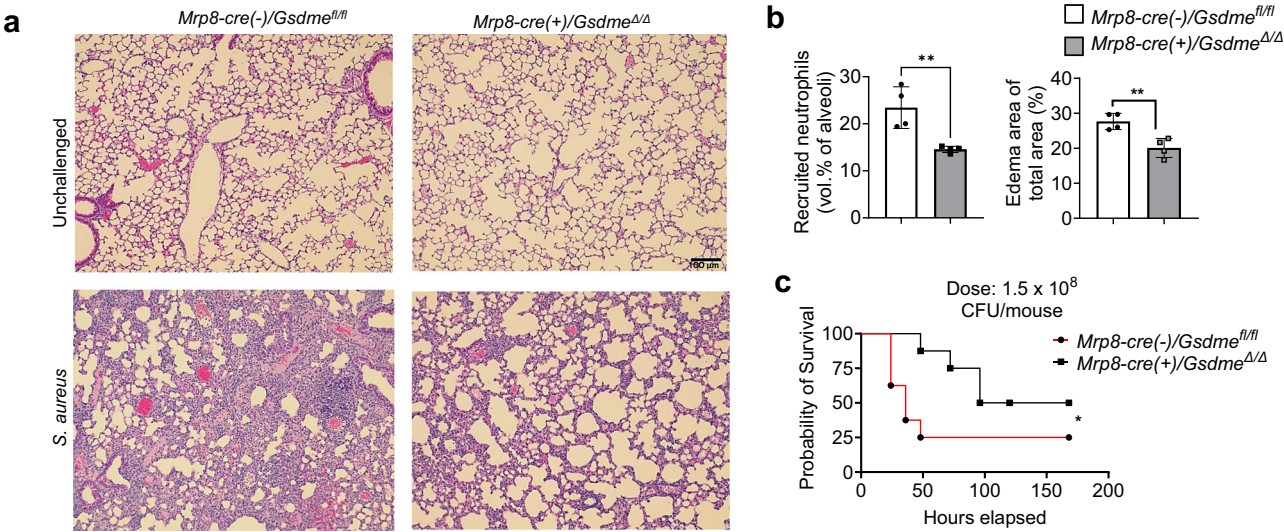

**Fig. 9 | GSDME disruption in neutrophils alleviated lung injury during *S. aureus*-induced pneumonia. a** Hematoxylin and eosin (H&E) stained lung sections from *Mrp8-cre(-)/Gsdme^{fl/fl}* and *Mrp8-cre(+)/Gsdme^{Δ/Δ}* mice pre and post *S. aureus*-induced pneumonia. The scale bar indicates 100 μm. Results represent a minimum of three biological replicates. **b** Assessment of pulmonary inflammation and damage via neutrophil recruitment (vol % of alveoli) and edema in the lung area. Data are represented as mean ± SEM (*n* = 4 mice/group). **P < 0.01 by two-sided Student's *t*-test. The data represents at least three independent experiments

(Recruited neutrophils, *P* = 0.0074; Edema area of total area, *P* = 0.0055). **c** Survival analysis of WT and neutrophil-specific *Gsdme* conditional KO post S. aureus infection. *Mrp8-cre(-)/Gsdme^{fl/fl}* and *Mrp8-cre(+)/Gsdme^{Δ/Δ}* mice (*n* = 8–9 mice/group) were intratracheally infected with *S. aureus* (1.5 × 10^8 CFUs/mouse) and monitored for survival over 7 days. Data were analyzed using the Gehan-Breslow-Wilcoxon test. **P < 0.05 (*P* = 0.0383). Source data are provided as a Source Data file.

compared with WT counterparts. There was decreased neutrophil accumulation in the lungs of *Gsdme* KO mice (Fig. S14b), which produced more anti-inflammatory IL-10 and less pro-inflammatory IL-6 than WT mice (Fig. S14c), resulting in an anti-inflammatory phenotype. As a result, KO mice showed significantly less lung injury, as evidenced by reduced pulmonary hyperemia (Fig. S14d) and total protein levels in BALF (Fig. S14e). Histopathological examination of lung sections after aspiration pneumonitis confirmed diminished neutrophil recruitment and edema formation in *Gsdme* KO mice (Fig. S14f–g). To confirm that the phenotype observed in KO mice was mediated by GSDME disruption in neutrophils, we also induced aspiration pneumonitis in *Mrp8-cre(+)/Gsdme^{Δ/Δ}* and *Mrp8-cre(-)/Gsdme^{fl/fl}* littermates (Fig. 10a). As observed in whole body KO mice, *Mrp8-cre(+)/Gsdme^{Δ/Δ}* mice also displayed decreased total leukocyte and neutrophil accumulation in the lungs (Fig. 10b), attenuated proinflammatory IL-6 and elevated anti-inflammatory Il-10 (Fig. 10c), and alleviated lung injury (Fig. 10d–g). These results confirm that neutrophil GSDME elicits a host inflammatory response in acid-induced aspiration pneumonitis and that genetic disruption of GSDME in neutrophils might alleviate acid-induced acute lung injury by regulating neutrophil death.

## Discussion

### A switch in the mode of cell death is an effective immune regulatory mechanism

Inflammation is an adaptive host response coordinated by sensors, mediators, and executors to noxious stimuli such as infection and tissue injury[59]. Regulated and controlled inflammation, for example in microbial infections, is considered beneficial, as it clears microbes with minimal damage to the host. However, dysregulated inflammation can be detrimental to the host as it leads to bystander tissue injury. Sometimes, the inflammation can cause more damage to the host than the pathogen, as observed in diseases such as pneumonia and sepsis[60]. Inflammation is initiated by tissue-resident macrophages, which sense microbes via their pattern recognition receptors (e.g., TLRs and NRLs) and produce mediators such as cytokines, chemokines, vasoactive amines, and eicosanoids. These recruit highly potent effector cells

such as neutrophils, which attempt to kill pathogens by releasing reactive oxygen species (ROS), reactive nitrogen species (RNS), proteases, and antimicrobial peptides[59–63]. Thus, host inflammatory responses can be regulated by interventions at various checkpoints in the inflammatory process[64], including sensors such as PAMP/DAMP receptors, mediators such as cytokines and chemokines, and executors such as immune cells.

In this study, we identified an immune regulatory mechanism. Both lytic and apoptotic programmed cell death remove senescent and damaged cells in living organisms, eliciting pro- and anti-inflammatory responses, respectively. However, it is unknown whether the host adopts specific death modes to modulate immune responses in different pathophysiological contexts. We discovered that the host inflammatory response can be modulated by manipulating the mode of neutrophil death. Neutrophil death is heterogenous under homeostatic conditions, comprising both pro-inflammatory lytic death and anti-inflammatory non-lytic apoptosis[13]. Here we found that GSDME was cleaved during neutrophil spontaneous death and that it controlled the mode of cell death. Both live cell imaging and FACS analysis revealed a shift in neutrophil death from lytic cell death to apoptosis, with the total count of healthy neutrophils remaining constant. The increase of the number of apoptotic neutrophils and consequently enhanced their efferocytotic clearance by macrophages, thereby augmenting anti-inflammatory responses, accelerating the resolution of inflammation, and attenuating tissue damage. To underscore the pathophysiological significance of this discovery, we implemented multiple disease models for validation purposes. These included HIEC-induced peritonitis, LPS-induced acute lung injury, Staphylococcus aureus pneumonia, and aspiration pneumonitis. In summary, our study sheds light on the manipulable nature of the host inflammatory response through the targeted alteration of the neutrophil death mechanism. With GSDME playing a crucial role in dictating cell death modes, strategic targeting of this protein opens avenues for modulating inflammatory responses during infection and inflammation, presenting potential therapeutic implications.

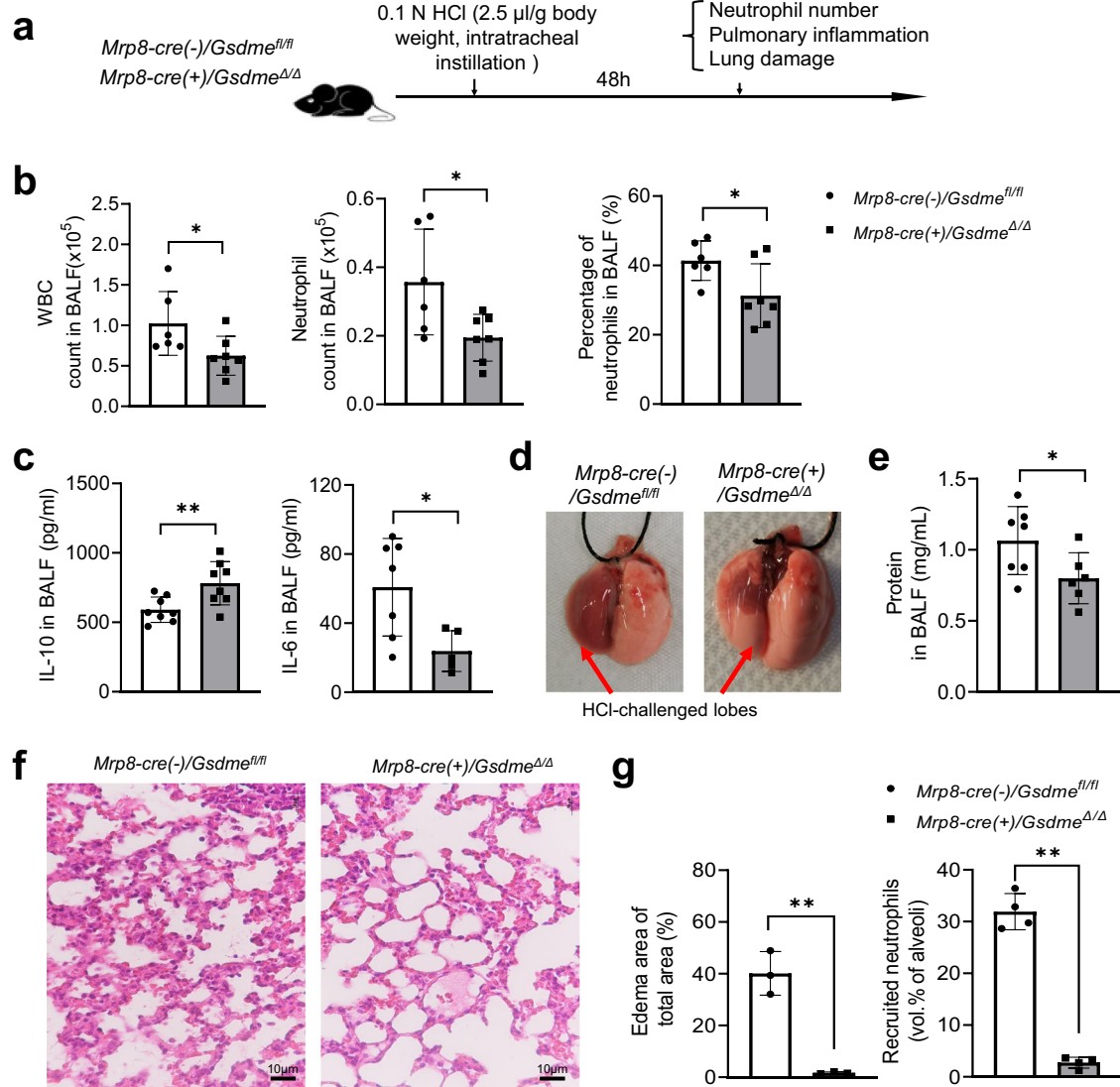

**Fig. 10 | Neutrophil-specific GSDME deletion attenuated inflammation and alleviated lung injury in acid aspiration pneumonitis. a** Experimental scheme for assessing acid-induced pulmonary inflammation and associated lung injury in WT ($Mrp8$-$cre(-)/Gsdme^{fl/fl}$) and neutrophil-specific $Gsdme$ KO ($Mrp8$-$cre(+)/Gsdme^{\Delta/\Delta}$) mice. Mice were challenged with 0.1 N HCl (2.5 µl/g body weight) and sacrificed after 48 h. **b** The number and percentage of neutrophils in BALF. WBC and neutrophil counts were determined by FACS using counting beads as described in Fig. S3. Cells were stained with APC-CD11b and PE-Ly6g antibodies. Neutrophils were identified as Ly6g$^+$CD11b$^+$ cells. All data are represented as mean ± SD, $n = 6$–7 mice, $^*P < 0.05$ by two-sided Student's $t$-test (WBC count, $P = 0.0469$; Neutrophil count, $P = 0.0283$; Percentage of neutrophils, $P = 0.0407$). **c** IL-10 and IL-6 levels in BALF of WT and neutrophil-specific $Gsdme$ KO mice were determined by enzyme-linked immunosorbent assay (ELISA). All data are represented as mean ± SD, $n = 7$–8 mice, $^*P < 0.05$, $^{**}P < 0.01$ by two-sided Student's $t$-test (IL-10, $P = 0.0099$; IL-6,

$P = 0.0213$). **d** Representative images of the lungs of acid-challenged WT and neutrophil-specific $Gsdme$ KO mice. The lung lobes challenged with HCl are indicated. Results are representative of at least three biological replicates. **e** BALF total protein level. Protein accumulation in inflamed lungs was measured using a protein assay kit. All data are represented as mean ± SD, $n = 6$–7 mice, $^*P < 0.05$ by two-sided Student's $t$-test ($P = 0.0478$). **f** Representative H&E-stained images of acid-challenged lung tissues. Results are representative of at least three biological replicates. **g** Neutrophil accumulation in alveoli was quantified as volume fraction of the alveolar space occupied by neutrophils. Pulmonary edema formation was quantified as the percentage of edema area in the total parenchymal region. Data are represented as mean ± SD of three experiments. $n \geq 6$ mice in each group. $^{**}P < 0.01$ by two-sided Student's $t$-test (Edema area of total area, $P = 0.0014$; Recruited neutrophils, $P = 0.0001$). Source data are provided as a Source Data file.

Neutrophil lytic death was primarily mediated by GSDME and was thus considered a form of pyroptosis in this study. Intriguingly, live cell imaging revealed that, while a substantial portion of pyroptotic neutrophils exhibited typical pyroptosis morphology including swollen membranes and condensed nuclei[65–67], others displayed normal multilobed or even diffused nuclei (Fig. 1b). To mitigate the impact of continuous culturing under a microscope on morphological analysis, we conducted morphological examinations at fixed time points without continuous monitoring. Even then, we observed neutrophils with atypical pyroptosis morphology (Fig. S3a). This suggests that the morphological alterations in neutrophils during pyroptosis may not

precisely mirror those observed in other cell types, such as macrophages.

**GSDMD and GSDME have distinct functions in neutrophil death**
GSDMD and GSDME are the only highly expressed gasdermins in neutrophils. The role of GSDMD in neutrophil death is well documented. Among gasdermin members, GSDMD is an established mediator of pyroptosis, an inflammatory mode of lytic cell death characterized by pore formation, plasma membrane rupture, and release of inflammatory cytokines including IL-1β. In macrophages, activation of canonical and non-canonical inflammasomes culminates

in the activation of caspases-1, −4, −5, −8, and −11, which cleave GSDMD to release the N-terminus domain responsible for pore formation in the plasma membrane and ultimately lytic cell death[28–31,68–71]. In neutrophils, activation of canonical caspase-1 inflammasomes[72,73] or non-canonical caspase-11 inflammasomes[38,74] also triggers GSDMD-mediated pyroptotic death, which is essential for host protection from extracellular or intracellular and cytosolic bacterial infections. Nevertheless, it is thought that, compared with macrophages, neutrophils are more resistant to GSDMD-mediated pyroptosis to preserve their viability for efficient microbial killing while maintaining GSDMD-dependent mechanisms for export of bioactive IL-1β[75–77]. Multiple studies have shown that canonical NLRC4/NLRP3-caspase-1 inflammasome signaling can induce GSDMD-dependent IL-1β release from neutrophils without pyroptotic death[38,74,78,79]. Low expression of SARM1, a key regulator of the rate of GSDMD pore formation at the plasma membrane, is a potential mechanism leading to neutrophil-specific resistance to pyroptosis[77,80]. GSDMD also plays a vital role in the generation of neutrophil extracellular traps (NETs)[38,39], which are extracellular DNA fibers networked with histones and cytoplasmic granule proteins that entangle and contain bacteria and that are also implicated in thrombosis, atherosclerosis, autoimmune diseases, and sepsis[81–83]. GSDMD-driven pyroptosis promotes PAD4-mediated histone citrullination and NET extrusion to protect against *Yersinia pseudotuberculosis* DyopM (Yptb DyopM) infection[73]. Silva et al. reported that GSDMD mediates sepsis by promoting NET formation and exacerbating inflammation in a mouse polymicrobial sepsis model[84]. However, while whole-body deletion of GSDMD protects against sepsis, its specific deletion from neutrophils not only fails to protect against sepsis but may even make it worse[85]. A recent report suggested that NET formation may be independent of GSDMD and pyroptotic cell death[86]. Another study showed that GSDMD drives canonical inflammasome-induced neutrophil pyroptosis but is dispensable for NETosis[72]. A small molecule inhibitor LDC7559, previously thought to block IL-1β release and NETosis by antagonizing GSDMD cytotoxicity[39], actually inhibits NETosis and subsequent tissue damage and inflammation via a GSDMD-independent mechanism - suppressing an excessive NOX2-dependent oxidative burst, a key step in executing classical NETosis[87].

In macrophages, GSDMD and GSDME appear to have overlapping and complementary functions in mediating pyroptosis. GSDMD-deficient macrophages cannot undergo pyroptosis and release IL-1β when briefly exposed to NLRP3 inflammasome activators. However, sustained inflammasome activation can alternatively trigger pyroptosis and IL-1β release in GSDMD-deficient macrophages by inducing caspase-8/−3 and GSDME cleavage[88]. GSDMD and GSDME are the two major gasdermins expressed by neutrophils. They both play a role in neutrophil spontaneous death but have distinct functions. GSDMD disruption in neutrophils increases the total number of heathy neutrophils[14] but did not alter the mode of neutrophil death (Fig. S7). Consistent with our results, another study also showed that GSDMD deletion did not alter the relative percentage of neutrophils undergoing apoptosis and lytic death[40]. By contrast, GSDME disruption completely abolished pyroptosis in neutrophils and skewed cell death towards apoptosis without affecting the total number of healthy neutrophils. Thus, GSDMD and GSDME have distinct and non-overlapping functions in neutrophil death (Fig. S15).

## The distinct functions of GSDMD and GSDME in neutrophil death are due to their different activation mechanisms

Gasdermins are cleaved by proteases to generate a cytotoxic N-terminus domain. In aging neutrophils, GSDME processing is initiated by proteinase 3 (PR3), a serine protease released from granules to the cytosol during neutrophil aging and that cleaves and activates caspase-3[8]. The resultant active caspase-3 in turn cleaved GSDME, leading to lytic neutrophil death (Fig. 3). It is noteworthy that neither

serine protease nor caspase-3 inhibitors completely blocked GSDME cleavage in aging neutrophils, indicating that GSDME cleavage may also be mediated by a caspase-3-independent mechanism. For instance, caspase 7 and Granzyme B (GrB) has also been shown to cleaved GSDME and, similarly, caspase-3 may also be activated by PR3-independent pathways such as canonical apoptotic signaling mediated by caspase-8 or −9 (Fig. S15). Conversely, in neutrophils, GSDMD can be cleaved by caspase-11[38], neutrophil elastase (ELANE)[14,39], and cathepsin G[40]. We previously reported that ELANE-induced cleavage of GSDMD in aging neutrophils leads to pyroptosis, thereby reducing neutrophil accumulation in the peritoneal cavity and hindering bacterial clearance during *E. coli* infection[14].

GSDME dictated the mode of neutrophil death, and its disruption did not alter the number of healthy neutrophils. By contrast, GSDMD contributed to both apoptosis and pyroptosis, and its disruption led to an overall increase in healthy neutrophil numbers but did not alter the mode of neutrophil death. This obvious difference in GSDMD and GSDME function can be explained by their different activation mechanisms (Fig. S15). PR3 is sequestered in granules in healthy neutrophils and is released during aging via lysosomal membrane permeabilization (LMP)[43] to trigger caspase-3 activation and GSDME-mediated pyroptosis. ELANE, another serine protease, is coincidentally released from granules to the cytosol where it cleaves and activates GSDMD[14,39]. However, a seminal study by Karmakar et al. showed that although N-GSDMD is required for IL-1β secretion in neutrophils, it does not form plasma membrane pores and thus does not directly induce cellular permeability nor pyroptosis. Instead, it predominantly localizes on and permeabilizes azurophilic granule membranes, amplifying LMP and further promoting leakage of neutrophil serine proteases into the cytosol, with PR3 mediating caspase-3 activation and ELANE mediating secondary cleavage of GSDMD[89]. GSDMD disruption shuts down GSDMD-mediated positive feedback regulation of LMP and thus suppresses LMP-mediated PR3 release from granules. This reduces caspase-3 activation, which in turn contributes to suppressing both apoptosis and pyroptosis and increasing healthy neutrophil numbers (Fig. S15). Of note, the elevated overall survival of GSDMD-deficient neutrophils was only observed at later time points (16 h in culture), again suggesting that GSDMD may indirectly regulate neutrophil spontaneous death. By contrast, GSDME acts downstream of caspase-3 and only contributes to pyroptosis without affecting caspase activation. Thus, GSDME disruption skews cell death towards apoptosis but does not alter healthy neutrophil numbers (Fig. S15).

## Neutrophil apoptosis and lytic death are parallel and independent events

For many cell types, when apoptotic cells are not scavenged, they progress to secondary lytic necrosis. A recent study identified GSDME as a central molecule regulating apoptotic cell disassembly and progression to secondary necrosis in 293 T cells and mouse bone marrow-derived macrophages[42]. Cells expressing GSDME progress to secondary necrosis when stimulated by apoptotic triggers but disassemble into small apoptotic bodies when GSDME is deleted. However, in neutrophils, lytic death is independent of apoptosis and not a result of secondary necrosis[13]. Consistently, elevated apoptosis triggered by GSDME disruption did not increase lytic neutrophil death at the early stage of neutrophil spontaneous death.

In another study, in the human T-lymphoblastic leukemia cell line CEM-C7, it was shown that the caspase-3-cleaved GSDME N-terminus can perforate the mitochondrial membrane, promoting cytochrome C release and apoptosome activation, thus forming a positive feedback loop to accelerate apoptosis[90]. Thus, GSDME disruption diminishes caspase-3 activation and apoptosis[90]. A similar mechanism was also identified in the central nervous system (CNS) where GSDME in neurons plays a role in mitochondrial damage and axon loss[91]. Again, this mechanism is unlikely to occur in neutrophils, whose mitochondrial

integrity is largely compromised during aging. In fact, GSDME disruption increased rather than suppressed caspase-3 cleavage (Fig. 2e), which was simply an indication of increased apoptotic cells in the GSDME-deficient neutrophil population.

## GSDME is a physiological pyroptosis inducer and a promising therapeutic target

*GSDME/DFNA5* was initially discovered as a gene associated with hearing loss[92]. Most research on GSDME has been in cancer, where GSDME acts as a potent tumor suppressor[28,41,93,94]. Cytotoxic chemotherapies induce GSDME-mediated pyroptosis via its cleavage by caspase-3, and GSDME-deficient mice are protected from chemotherapy-induced tissue damage[28]. GSDME expression is low in many cancers. In mice, ectopic GSDME expression in *Gsdme*-repressed tumors inhibited tumor growth mediated by Granzyme B in killer cytotoxic lymphocytes, which cleaved and activated GSDME in target cancer cells, leading to caspase-independent pyroptosis[41]. GSDME also induced pyroptosis of tumor cells during oncolytic viral cancer therapy: oncolytic parapoxvirus ovis reduced ubiquitination of GSDME and stabilized it to initiate pyroptosis of tumor cells expressing low levels of GSDME[95]. Interestingly, a recent study also reported a tumor promoting role for GSDME in pancreatic ductal adenocarcinoma (PDAC) cells expressing high levels of GSDME to mediate resistance to pancreatic enzymatic digestion through a GSDME–YBX1–mucin axis[96]. Thus, GSDME can be both beneficial[96] and detrimental[41,93,94] to tumor growth. GSDME has also been reported to play a role in chimeric antigen receptor (CAR) T cell therapy in cancer patients: Granzyme B released by CAR-T cells activated caspase 3 and induced GSDME-mediated pyroptosis in macrophages, triggering a serious complication known as cytokine release syndrome[97].

GSDME has also been implicated in pathogen infections, and it was shown to initiate a lethal cytokine storm in *H7N9* avian influenza viral infection[98]. GSDME deficiency reduces disease severity of enteroviral 71 infection[99]. Zika virus also causes GSDME-mediated pyroptosis in placental cells, leading to adverse fetal outcomes. In this context, RIG-1 senses genomic viral RNA to induce TNF-α secretion that leads to caspase −8 and −3 activation via the extrinsic apoptotic pathway. The cleaved caspase-3 then induces pyroptosis by cleaving GSDME in placental trophoblasts[100]. Conversely, GSDME is also essential for host defenses against various pathogens. For instance, GSDME activation is required for neutrophil lysis upon *Yersinia*-induced apoptotic caspase activation[37]. GSDME, cleaved and activated by caspase-3, induces neutrophil pyroptosis and thus facilitates production of IL-1β, which is essential for protection against *Yersinia* infection, with *Gsdme* KO mice more susceptible compared to wild-type counterparts[37]. Additionally, GSDME pore formation in TH17 cells was identified as a mechanism of unconventional IL-1α release, contributing to an antifungal host defense response[101].

Here we uncovered an important function for GSDME in the resolution of inflammation. By dictating the mode of neutrophil death, GSDME can be therapeutically targeted to modulate host defenses and inflammation.

## Methods

### Animals

All mouse work was performed in accordance with protocols approved by the Institutional Animal Care and Use Committee at Boston Children's Hospital (BCH) and the Institute of Hematology Chinese Academy of Medical Sciences (IHCAM). All necessary precautions were followed to minimize pain and stress during experimental procedures. CO2 was used as a humane method of euthanasia for all mouse work. *C57BL/6 J* wild-type (WT, Strain #:000664), *B6.Cg-Tg(Mrp8-cre,-EGFP)1Ilw/J* (Strain #:021614), and *C57BL/6-Tg(CAG-EGFP)1O sb/J (B6 ACTb-EGFP)* (Strain #:003291) mice were purchased from Jackson Laboratories (Bar Harbor, ME). Mice were kept in a

specific pathogen-free cage system under a 12/12 h light/dark cycle with free access to food and water. *Gsdmd* KO mice were generated as previously described[14,85]. *Gsdme* KO mice were generated on a pure *C57BL/6 J* background using the CRISPR/Cas9 system targeting exon 3 (Fig. S2). *Gsdme* deletion was confirmed using standard genotyping and immunoblotting. *Gsdme*^flox/flox^ mice were produced by inserting loxP into both ends of exon 3 (Fig. S11). Neutrophil-specific conditional *Gsdme* KO mice (*Mrp8-cre(+)/Gsdme*^Δ/Δ^) were generated by crossing *Gsdme*^flox/flox^ mice with *B6.Cg-Tg(Mrp8-cre,-EGFP)1Ilw/J* mice (Fig. S11). Heterozygous F1 parents were used for mating to generate *Mrp8-cre(+)/Gsdme*^Δ/Δ^ and *Mrp8-cre(-)/Gsdme*^f/f^ littermates.

### Imaging-based analysis of neutrophil death

Bone marrow (BM) neutrophils (from 8–12 weeks old male mice) were isolated (Supplementary Methods) and cultured in RPMI 1640 supplemented with 20% heat-inactivated fetal bovine serum (FBS, Gibco, Thermo Fisher Scientific, Waltham, MA) and penicillin-streptomycin at a density of $2 \times 10^6$ cells/ml at 37 °C in a 5% $CO_2$ incubator. At each indicated time point, the total cell number was counted using a hemocytometer, and this number was defined as the "remaining cell number". "Disappeared cell number" was calculated as the total cell number at 0 h minus the remaining cell number. To accurately characterize the neutrophil death pattern, PI and FITC-Annexin V were added into the culture medium and mixed by gentle tilting without pipetting to avoid destroying the lytic cell death. Images were taken under a Nikon Ti Eclipse inverted microscope. At least four independent fields were imaged using a 40x objective. At the indicated time points, the numbers of total and swollen cells per field were counted in brightfield images. The numbers of PI+ or Annexin V+ cells per field were counted in fluorescent imaging fields. The percentage of swollen cells was calculated as the ratio of swollen cells to total cells. The total swollen cell number was calculated as the product of its percentage and "remaining cell number" at the same time point. The percentage and number of PI+ or Annexin V+ cells were calculated in the same way. At least 200 cells were counted to calculate the percentage of each cell population. For morphological analysis, bone marrow neutrophils were seeded in a confocal dish (NEST Scientific). Neutrophils were labeled with APC-Ly6g (BioLegend, #108423, Clone 1A8), and Sytox Orange dye (0.25 μM, Thermo Fisher Scientific) and FITC-Annexin V were used to assess neutrophil death at 4 h and 20 h. Images were captured using a confocal microscope (UltraVIEW VOX, PerkinElmer) with a 60x objective.

### Time-lapse imaging of neutrophil death

To track neutrophil death in real time, neutrophils isolated from 8–12 weeks old male mice, were seeded ($2 \times 10^4$ cells in 10 μl) in a micro-insert (ibidi GmbH, Gräfelfing, Germany) placed in a 20 mm glass-bottomed confocal dish (NEST Scientific, Jiangsu, China). PI (2 μg/ml) and FITC-Annexin V (2.5 μg/ml) were added into the culture medium to stain dying cells. Time-lapse imaging was performed using a Delta Vision Ultra inverted microscope equipped with an incubator box to maintain $CO_2$ levels (5%), temperature (37 °C), and humidity (95%). Brightfield and fluorescence field images were acquired every 5 min for 15 h using a 60x oil objective. Images were processed using Volocity (Quorum Technologies, Puslinch, ON) and ImageJ (NIH) software. By tracking cells in time-lapse videos, the fate and ultimate mode of death of each cell were determined based on both morphological changes and AV/PI staining (Fig. 1b). The absolute number of each cell population was calculated as the product of the initial cell count (time 0) times its percentage at each time point. To calculate the percentages, at least 200 cells were tracked.

### Flow cytometry-based analysis of neutrophil death

Isolated neutrophils were cultured as described above. At each indicated time point, cells were harvested, stained with Annexin V-FITC

and PI, and analyzed via flow cytometry. For accurate measurement of healthy and dying neutrophil numbers in the culture, we used flow cytometry-based counting beads (CountBright™ Plus Absolute Counting Beads; Thermo Fisher Scientific) to count the absolute numbers of each neutrophil population. Samples were run on a BD FACSCanto II flow cytometer and analyzed using FlowJo software (BD Biosciences). Annexin V and PI double-negative cells defined healthy cells.

### Efferocytosis in peritonitis and pneumonia

Isolated BM neutrophils from 8–12 weeks old male WT or *Gsdme* KO mice were cultured for 20 h and stained with 1 μM CFSE for 10 min. After washing, $5 \times 10^6$ CFSE-labeled intact neutrophils were intraperitoneally or intratracheally injected into recipient mice challenged with LPS (5 mg/kg body weight) or heat-inactivated *E. coli* (ATCC19138, $10^7$ CFU) for 48 h. Peritoneal fluid or BALF was harvested 1 h after cell injection. Cells in lavage fluid were collected and resuspended in 100 μL ice-cold PBS. Cells were then incubated with TruStain FcX™ (anti-mouse CD16/32) Fc blocking antibody at 4 °C for 20 min and stained with APC-Ly6g, Percp-Cy5.5-F4/80 (BioLegend, #157318, Clone BM8), PE-Cy7-CD11b (BioLegend, #101216, Clone M1/70), or APC-Cy7-Ly6c (BioLegend, #128026, Clone HK1.4) at 4 °C for 20 min. The percentage of CFSE$^+$ cells in macrophages and the percentage of Ly6g$^-$CFSE$^+$ cells in total CFSE$^+$ cells were analyzed by flow cytometry using a BD FACSCanto II flow cytometer. In another setup, fresh neutrophils were labeled with CFSE and were intraperitoneally or intratracheally injected into recipient mice immediately. The efferocytosis of aging neutrophils by macrophages was measured as described above 6 h after injection.

### Statistical analysis

For most experiments, statistical significance was determined between two groups using the two-tailed, unpaired, Student's *t*-test. Data are presented as means ± SD. A *p*-value ≤ 0.05 was considered statistically significant. All statistical tests were performed and graphics were created in GraphPad Prism 9 (GraphPad Software, La Jolla, CA). Unless specifically mentioned, experiments were performed independently at least three times and the data were pooled and analyzed together.

### Reporting summary

Further information on research design is available in the Nature Portfolio Reporting Summary linked to this article.

## Data availability

All sequencing data have been deposited at NCBI GEO depository under accession number: GSE137540. The data generated and/or analyzed during the current study are available from the corresponding authors. Source data are provided with this paper.

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

## Acknowledgements

The authors thank Judy Lieberman, Yi Zheng, Jose Cancelas-Perez, John Manis, Li Chai and Shin-young Park for helpful discussions and suggestions. Fengxia Ma was supported by the grants from Chinese Academy of Medical Sciences (CAMS) Innovation Fund for Medical Sciences (2021-I2M-1-040, 2022-I2M-JB-015), National Natural Sciences Foundation of China (82171756), Haihe Laboratory of Cell Ecosystem Innovation Fund) (22HHXBSS00019), Natural Science Foundation of Tianjin City (22JCYBJC01120). Fei Liu was supported by Tianjin Municipal Science and Technology Commission Grant (21JCQNJC01550). Hongbo R Luo was supported by National Institutes of Health grants 1R01AI142642, 1R01AI145274, 1R01AI141386, R01HL092020, and P01HL158688. L.G. and A.H. were supported by NIH training grant T32HL066987.

## Author contributions

H.R.L. and F.M. inputted conceptualization; H.R.L., F.M., L.G., Q.R., Y.F., A.B., A.H., and H.Y. designed the methods; L.G., Q.R., Y.F., A.B., A.H., T.C and R.X acquired samples; F.M., L.G., Q.R., Y.F., A.B., A.H., F.L., X.X., T.C. and H.Y. performed analysis.; H.R.L., F.M., and H.Y. provided resources; H.R.L. supervised all work. H.R.L, F.M. and L.G. wrote the paper with input from all the authors. All coauthors read, reviewed, and approved the manuscript.

## Competing interests

The authors declare no competing interests.
