## [Peer Review File · Nature Communications]

Gasdermin E dictates inflammatory responses by controlling the mode of neutrophil deathREVIEWER COMMENTS

Reviewer #1 (Remarks to the Author):

General Comments: This MS describes physiologically significant and mechanistically novel roles for neutrophil Gasdermin E (GSDME) in regulating the mode of neutrophil death between pro-inflammatory lytic pyroptosis and anti-inflammatory apoptosis. This team of investigators have extensive prior experience in characterizing the constitutive or spontaneous cell death signaling pathways that underlie the well-established short-lifespan and rapid turnover of neutrophils. The study design incorporates both ex vivo experiments with isolated primary murine neutrophils (bone marrow or elicited peritoneal) and three distinct in vivo models of neutrophil-mediated inflammation: 1) peritonitis induced by injection of heat-killed E coli; 2) lung inflammation induced by intra-tracheal injection of LPS; 3) sterile lung injury induced by intra-tracheal injection of acid (HCl). The design also utilizes mice with global genetic deletion of GSDME, as well as mice with neutrophil-specific GSDME deletion. The major findings are: 1) GSDME is cleaved and activated during constitutive neutrophil death via signaling axis involving leakage of the proteinase 3 (PR3) serine protease from azurophilic granules into the cytosol to drive PR3-mediated cleavage of caspase-3, casp3-cleavage of GSDME, accumulation of pore-forming GSDME cleavage products in the plasma membrane, and consequent lytic pyroptotic death. 2) In the absence of GSDME, this cascade is skewed to result in enhanced accumulation of "conventional" apoptotic neutrophils that are rapidly cleared by macrophage-mediated efferocytosis in the in vivo inflammation model. 3) This enhanced efferocytosis of apoptotic GSDME-deficient neutrophils licenses the well-established anti-inflammatory polarization of the efferocytosing macrophages (reduced production of IL-6 and IL-1b versus increased IL-10 secretion) to attenuate the feed-forward cycle of increased tissue damage by recruited neutrophils driven to a pro-inflammatory phenotype.

In general, the experimental design is comprehensive and the data interpretations are buttressed by appropriate positive and negative control studies. A few technical and textual issues require clarification or correction. Additionally, the breadth and depth of the study would be further enhanced by inclusion of two additional sets of experiments with 1) human neutrophils and 2) an in vivo mouse infection model.

Specific Comments:

1. It would be informative to include descriptive analysis of whether the PR3> caspase-3> GSDME> pyroptotic cascade is also operative in isolated human blood neutrophils incubated ex vivo under conditions that facilitate constitutive death.
2. The three in vivo inflammation models in the current MS involve either explicitly sterile (acid inhalation) or de facto sterile but infection-related (intraperitoneal heat-killed bacteria and intratracheal LPS) challenges. Although an extensive analysis of how neutrophil GSDME modulates their anti-bacterial functions is beyond the scope of the study, it would be informative to include at least one model of in vivo bacterial infection. The authors have recently reported (ref 75) how the acute septic responses of mice to cecal ligation and perforation (CLP) challenge are very distinct in animals with global GSDMD deficiency versus neutrophil-specific GSDMD knockout. It should be straightforward to adapt this acute model (12 hrs for inflammatory markers; 3 days for survival) to analysis with the GSDME-deficient mouse strains already in hand.
3. Results, page 7, paragraph 3 and Supplemental Methods, page 38, last paragraph (and related figure S5): These sections and the figure describe the use of "GFP-expressing recipient mice" or "GFP-positive recipient mice". The exact nature of these GFP-expressing mice is unclear (to me). The only GFP expressing strain noted in Methods (page 17) is the transgenic B6.Cg-Tg (Mrp8-cre-EGFP) 1 Ilw/J. The authors should clarify whether these mice or another strain were used as the recipients.
4. Methods, page 17, second paragraph: "Isolated neutrophils were cultured as described above." There is no previous description of this in the preceding paragraph. Rather, the isolation and incubation is described in Supplemental Methods, pages 36-37.

5. Methods, pages 17-18, "Imaging-based analyses...", "Time-lapse imaging...", "Flow cytometry based analyses...": These ex vivo methods only indicate use of "isolated neutrophils". I assume these are isolated bone marrow neutrophils, but this should be specified. Also, the incubation medium used for these ex vivo studies should be specified, e.g. plus or minus serum.
6. Supplemental Figure 1 legend: The legend (and also relevant Methods section) should provide more methodological details on these scRNA analyses. E.g. were only bone marrow neutrophils used or were blood neutrophils also analyzed? Also, the abbreviations in the Figure should be defined in the legend.
7. Figs S4A, S10E, S11F, 1A, 1B, 7F: The micrograph images need scale bars.
8. Figure 5A, 5H legends, page 33 and Supplemental Methods, page 38 use the terms "after infection" or "post-infection" to describe some of the LPS or HIEC challenges. Since no living bacteria were used the terms "infection" should be replaced by "LPS or HIEC challenges".
9. Page 3, paragraph 3, line 1: "identified humans" > "identified in humans"
10. Page 39, line BALF collection: "as preciously described" > "as previously described"

Reviewer #2 (Remarks to the Author):

The manuscript delineates the implications of programmed cell death regulation, encompassing both apoptosis and pyroptosis (lytic cell death), in the context of neutrophil aging and inflammatory responses associated with diverse pathologies. While acknowledging the established role of Gasdermin E (GSDME) in pyroptosis regulation within neutrophils, the current study unveils showed that the absence of GSDME leads to a reduction in pyroptosis alongside an increase in apoptosis, ultimately culminating in an anti-inflammatory phenotype. This anti-inflammatory effect of apoptotic neutrophils is attributed to their facilitation of macrophage efferocytosis. To ascertain the in vivo implications of neutrophil-specific GSDME, a novel mouse model integrating *Mrp8-cre* and *Gsdme fl/fl* components is devised. This platform is subsequently employed to probe the role of neutrophil GSDME in disease models encompassing peritonitis induced by heat-inactivated *E. coli* (HIEC), acute lung injury provoked by lipopolysaccharide (LPS), and lung inflammation incited by hydrochloric acid (HCl). However, several of these studies yield repetitive outcomes, warranting a thorough reassessment of their scientific implications.

Major comment

1. The decreased lytic cell death (pyroptosis) observed in GSDME-deficient neutrophils, as depicted in Figure 1D of the present manuscript, aligns with the results from Figure 3A and 3C of the reference provided: ("RIPK1 activates distinct gasdermins in macrophages and neutrophils upon pathogen blockade of innate immune signaling." Proceedings of the National Academy of Sciences of the United States of America vol. 118,28 (2021): e2101189118. doi:10.1073/pnas.2101189118).
 2. The GSDME-caspase3 pathway depicted in Figure 3 of the manuscript has been previously addressed in the referenced article (pnas.2101189118). Hence, it is crucial to elucidate the differentiation inherent in the initial part of this manuscript compared to earlier studies.
 3. The data illustrated in Figures 2A-D exhibits a degree of redundancy with the information already delineated in Figure 1.
 4. In Figures 1-3, neutrophils derived from GSDME-KO and WT sources underwent a 24-hour culture devoid of specific stimulation to discern variations in the pyroptosis-to-apoptosis ratio. However, the subsequent in vivo investigations depicted in Figures 4-7 primarily involve assessing apoptosis-efferocytosis dynamics or gauging inflammation levels within artificially induced animal models, rather than under homeostatic conditions.
- To bridge the gap between the in vitro results presented in Figures 1-3 and the in vivo results depicted in Figures 4-7, it is proposed to incorporate additional data demonstrating apoptosis and pyroptosis after stimulation of neutrophils in an in vitro experimental setting with proinflammatory

cytokines that mimic *in vivo* inflammation.

5. In relation to the disease scoring performed within the HIEC-peritonitis model, as depicted in Figure 4G-L, it appears that relying solely on the fluctuating patterns of IL-10 and IL-1 β cytokines for gauging the inflammation degree might introduce subjectivity. To enhance the robustness of the evaluation, it would be beneficial to include additional disease scoring techniques and present the concurrent changes in various anti- and pro-inflammatory cytokine levels.

6. Figure 5-7 seem to convey comparable narratives in parallel, despite the variations in disease models. To enhance the analytical depth, introducing a mechanistic investigation that is currently absent could be considered.

7. Furthermore, considering the availability of a Mrp8-cre x Gsdme fl/fl mouse system, it would be advantageous to replace the data obtained from the whole-body knockout system with the conditional knockout system.

Reviewer #3 (Remarks to the Author):

In this manuscript, Ma et al. found that neutrophils spontaneously died by GSDME-mediated pyroptosis. This GSDME activation was mediated by proteinase-3 and subsequent caspase-3 activation. GSDME deficiency did not alter the neutrophil survival rate, but it switched a form of RCD from pyroptosis towards apoptosis. To assess clinical relevance, the authors conducted LPS-induced lung inflammation and acid aspiration lung injury models in GSDME-KO (systemic) and neutrophil-specific GSDME-KO (using Mrp8-Cre) mice, and showed that neutrophil GSDME contributes to the development of lung injury in both mouse models. This study is well performed and the results are potentially interesting. However, there are many concerns that should be addressed.

1. The most important finding is that neutrophils undergo spontaneous lytic death (pyroptosis) by spontaneous cleavage of GSDME. However, it is generally believed that apoptosis is a typical form of RCD in spontaneous neutrophil death and apoptotic neutrophils are cleared by efferocytosis. Therefore, this reviewer wonders whether spontaneous pyroptosis could be artificial and really occur in physiological *in vivo* situations. The authors should provide the data on spontaneous pyroptotic neutrophils (with GSDME cleavage) escaping from efferocytosis in mice.

2. Another main concern is that the findings of this study are obtained only in mice, but not humans. Therefore, the translational value of the reported findings is difficult to assess. The authors should investigate the occurrence of spontaneous lytic neutrophil death and the role of GSDME in its process.

3. The authors showed that spontaneous apoptosis was promoted in GSDME-deficient neutrophils by living cell image and flow cytometry analyses. However, several inconsistent data between these analyses make obscure the conclusion. Although the percentage of apoptotic cells evaluated by AV+PI⁻ cells was increased in GSDME-deficient neutrophils in early time points, it was unchanged in later time points. Furthermore, AV+PI⁺ cells including pyroptosis and secondary necrosis were increased or unchanged in GSDME-deficient neutrophils or unchanged. These results might reflect that GSDME deficiency enhanced caspase-3 activation and accelerated apoptosis in early time points. The present results are not sufficient to conclude that GSDME deficiency skews cell death from lytic to apoptotic cells.

4. Although the authors discussed the distinct functions of GSDMD and GSDME in neutrophil spontaneous death, the role of GSDMD was not fully assessed. Whereas the number of AV⁻PI⁻ live cells was increased in GSDMD-deficient neutrophils, the number of dead cells was not shown. The number of AV+PI⁻ neutrophils and AV+PI⁺ neutrophils must be increased in GSDMD-deficient neutrophils. Furthermore, the cleavage of GSDMD during neutrophil spontaneous death should be analyzed.

Technical concerns

5. Although the authors used PI staining to evaluate the dead cells in live cell imaging, PI can be

entered in living cells in a longer culture period. Other cell-impermeable dyes, such as SYTOX Orange/DeepRed should be tested.

6. Some images of lytic cell death (Fig. 1A and B) are different from typical images of pyroptosis (swollen membrane and condensed nuclei). Morphological analysis should be validated at a fixed time point because culture under a microscope might affect viability.

7. The authors evaluated pyroptosis or lytic cell death by morphological change and PI staining. Other methods such as LDH release should be tested.

We appreciate the conceptual recognition of the importance of our work by the reviewers and are pleased with the conclusions that " the experimental design is comprehensive and the data interpretations are buttressed by appropriate positive and negative control studies" (Reviewer #1), " The manuscript delineates the implications of programmed cell death regulation, encompassing both apoptosis and pyroptosis" (Reviewer #2), " This study is well performed and the results are potentially interesting" (Reviewer #3). We thank the reviewers for raising insightful comments that helped us improve the study and the manuscript substantially. We revised our manuscript by closely following these suggestions. We performed additional experiments/analyses in the past 3 months and added a significant number of new results (Fig.1h, Fig.3e, Fig.4k, n, o, Fig.7, Fig.S3, Fig.S6, Fig.S7a, c, d, and Fig.S8) to address the concerns raised by the reviewers. Specific responses are as follows (changes/new sections are marked **red** in the revised manuscript):

Reviewer #1

1. It would be informative to include descriptive analysis of whether the PR3> caspase-3> GSDME> pyroptotic cascade is also operative in isolated human blood neutrophils incubated ex vivo under conditions that facilitate constitutive death.

Response: We conducted the experiment as suggested. Indeed, the PR3> caspase-3> GSDME> pyroptotic cascade was also operative in human neutrophils. We have included this result in Fig.S8 in the revised manuscript (described on page 8).

2. The three in vivo inflammation models in the current MS involve either explicitly sterile (acid inhalation) or de facto sterile but infection-related (intraperitoneal heat-killed bacteria and intratracheal LPS) challenges. Although an extensive analysis of how neutrophil GSDME modulates their anti-bacterial functions is beyond the scope of the study, it would be informative to include at least one model of in vivo bacterial infection. The authors have recently reported (ref 75) how the acute septic responses of mice to cecal ligation and perforation (CLP) challenge are very distinct in animals with global GSDMD deficiency versus neutrophil-specific GSDMD knockout. It should be straightforward to adapt this acute model (12 hrs for inflammatory markers; 3 days for survival) to analysis with the GSDME-deficient mouse strains already in hand.

Response: This is a great suggestion. We have now included a model of in vivo bacterial infection (Fig.7). Since we focused on inflammation and tissue damage in the lungs, we chose to use a Staphylococcus aureus (gram-positive bacteria)-induced pneumonia model. The new results not only provide insight into how neutrophil GSDME modulates their anti-bacterial function, but also demonstrate that the GSDME-mediated immune modulation is a general mechanism that extends beyond LPS-elicited inflammation (described on page 12).

3. Results, page 7, paragraph 3 and Supplemental Methods, page 38, last paragraph (and related figure S5): These sections and the figure describe the use of "GFP-expressing recipient mice" or "GFP-positive recipient mice". The exact nature of these GFP-expressing mice is unclear (to me). The only GFP expressing strain noted in Methods (page 17) is the transgenic B6.Cg-Tg (Mrp8-cre-EGFP) 1 Ilw/J. The authors should clarify whether these mice or an another strain were used as the recipients.

Response: Sorry for the omission. The "GFP-expressing mice" refers to C57BL/6-Tg(CAG-EGFP)10sb/J (B6 ACTb-EGFP) mice. This transgenic mouse line, with an "enhanced" GFP (EGFP) cDNA under the control of a chicken beta-actin promoter and cytomegalovirus enhancer, have widespread EGFP fluorescence, with the exception of erythrocytes and hair. We have added this information in the revised manuscript (pages 8 and 20).

4. Methods, page 17, second paragraph: "Isolated neutrophils were cultured as described above." There is no previous description of this in the preceding paragraph. Rather, the isolation and incubation is described in Supplemental Methods, pages 36-37 (page 20 in the revised manuscript).

Response: Sorry for the confusion. We modified this sentence as suggested.

5. Methods, pages 17-18, “Imaging-based analyses...”, “Time-lapse imaging...”, “Flow cytometry based analyses...”: These ex vivo methods only indicate use of “isolated neutrophils”. I assume these are isolated bone marrow neutrophils, but this should be specified. Also, the incubation medium used for these ex vivo studies should be specified, e.g. plus or minus serum.

Response: We added these details in the revised manuscript (page 20 in the revised manuscript).

6. Supplemental Figure 1 legend: The legend (and also relevant Methods section) should provide more methodological details on these scRNA analyses. E.g. were only bone marrow neutrophils used or were blood neutrophils also analyzed? Also, the abbreviations in the Figure should be defined in the legend.

Response: As suggested, we provided more methodological details on the scRNA analyses (Fig.S1 legend).

7. Figs S4A, S10E, S11F, 1A, 1B, 7F: The micrograph images need scale bars.

Response: We added scale bars as suggested (F1a, b, F8, FS5, FS13, and FS14 in the revised manuscript).

8. Figure 5A, 5H legends, page 33 and Supplemental Methods, page 38 use the terms “after infection” or “post-infection” to describe some of the LPS or HIEC challenges. Since no living bacteria were used the terms “infection” should be replaced by “LPS or HIEC challenges”.

9. Page 3, paragraph 3, line 1: “identified humans” > “identified in humans”

10. Page 39, line BALF collection: “as preciously described” > as previously described”

Response: Thanks for pointing this out. We modified the related sentences as suggested.

Reviewer #2

1. The decreased lytic cell death (pyroptosis) observed in GSDME-deficient neutrophils, as depicted in Figure 1D of the present manuscript, aligns with the results from Figure 3A and 3C of the reference provided: (“RIPK1 activates distinct gasdermins in macrophages and neutrophils upon pathogen blockade of innate immune signaling.” Proceedings of the National Academy of Sciences of the United States of America vol. 118,28 (2021): e2101189118. doi:10.1073/pnas.2101189118).

Response: Thank you for your comment. Chen et al. elegantly demonstrated that GSDME, but not GSDMD, drives neutrophil lysis following activation of extrinsic and intrinsic apoptosis. Indeed, our observation of decreased lytic cell death (pyroptosis) in GSDME-deficient neutrophils aligns nicely with these results. However, our current study primarily investigates the role of GSDMD/E in spontaneous neutrophil death and during the resolution of inflammation—a domain previously unexplored. We’ve pinpointed GSDME as the central regulator that determines the nature of neutrophil death, whether lytic or apoptotic. Importantly, by controlling the mode of neutrophil death, GSDME dictates host inflammatory outcomes, providing a novel therapeutic target for infectious and inflammatory diseases. We have integrated this discussion into the revised manuscript on page 7.

2. The GSDME-caspase3 pathway depicted in Figure 3 of the manuscript has been previously addressed in the referenced article (pnas.2101189118). Hence, it is crucial to elucidate the differentiation inherent in the initial part of this manuscript compared to earlier studies.

Response: Chen et al. (pnas.2101189118) demonstrated that both extrinsic and intrinsic apoptosis pathways activate caspase-3 and GSDME in neutrophils. Upon infection with Yersinia, RIPK1 promotes caspase-3–dependent GSDME activation, leading to neutrophil pyroptosis. The caspase inhibitor Q-VD-Oph (QVD)

inhibits both caspase-3 and GSDME activation in neutrophils. In our current study, we focus on the role of GSDME in neutrophil spontaneous death and the resolution of inflammation. We validated the PR3> caspase-3> GSDME> pyroptotic cascade in this context. We have incorporated this discussion in the revised manuscript (pages 7-8).

3. The data illustrated in Figures 2A-D exhibits a degree of redundancy with the information already delineated in Figure 1.

Response: Indeed, both Figure 1 and Figure 2 illustrate that disruption of GSDME shifts neutrophil death towards apoptosis. This observation serves as the cornerstone of our current study. We believe it is essential to draw conclusions using different approaches: living cell imaging-based analysis in Figure 1 and FACS-based analysis in Figure 2. If the reviewer feels it's necessary, we can certainly move Figure 2 to the SUPPLEMENTAL INFORMATION section.

4. In Figures 1-3, neutrophils derived from GSDME-KO and WT sources underwent a 24-hour culture devoid of specific stimulation to discern variations in the pyroptosis-to-apoptosis ratio. However, the subsequent in vivo investigations depicted in Figures 4-7 primarily involve assessing apoptosis-efferocytosis dynamics or gauging inflammation levels within artificially induced animal models, rather than under homeostatic conditions. To bridge the gap between the in vitro results presented in Figures 1-3 and the in vivo results depicted in Figures 4-7, it is proposed to incorporate additional data demonstrating apoptosis and pyroptosis after stimulation of neutrophils in an in vitro experimental setting with proinflammatory cytokines that mimic in vivo inflammation.

Response: Thanks for the insightful suggestion. We've carried out the experiments as recommended. The results indicate that proinflammatory cytokines IL-1 β and IL-6 had no significant impact on neutrophil death. As previously reported, TNF- α triggered significant apoptosis. Importantly, disruption of GSDME consistently shifted neutrophil death towards apoptosis, even in the presence of various proinflammatory cytokines. We have included this result in Fig.S6 in the revised manuscript (described on pages 6-7).

5. In relation to the disease scoring performed within the HIEC-peritonitis model, as depicted in Figure 4G-L, it appears that relying solely on the fluctuating patterns of IL-10 and IL-1b cytokines for gauging the inflammation degree might introduce subjectivity. To enhance the robustness of the evaluation, it would be beneficial to include additional disease scoring techniques and present the concurrent changes in various anti- and pro-inflammatory cytokine levels.

Response: Based on the reviewer's suggestion, we conducted two additional assays to assess the extent of inflammation: the BALF total protein level and the MPO level. The results of these assays are presented in Fig.4n and o (described on page 10).

6. Figure 5-7 seem to convey comparable narratives in parallel, despite the variations in disease models. To enhance the analytical depth, introducing a mechanistic investigation that is currently absent could be considered.

Response:

- Both lytic and apoptotic programmed cell death remove senescent and damaged cells in living organisms, eliciting pro- and anti-inflammatory responses, respectively. However, it is unknown whether the host adopts specific death modes to modulate immune responses in different pathophysiological contexts. We feel our conclusion that GSDME dictates inflammatory responses by controlling the mode of neutrophil death is significant. Thus, to validate its pathophysiological relevance, we used multiple disease models. The experiments presented in Figures 5-7 definitively demonstrate that neutrophil-specific deletion of

GSDME facilitates the resolution of inflammation and alleviates tissue injury. We further emphasized the significance of these in vivo studies in the revised manuscript (page 14).

- As the reviewer correctly pointed out, it is important to validate the underlying mechanisms in animal models. There are two major novel mechanistic findings reported here:
 1. **GSDME controls the mode of neutrophil death.** In the revised manuscript, we incorporated an additional experiment to examine the nature of neutrophil death in vivo (Fig.4k). This experiment confirmed that GSDME disruption does indeed elicit an increase in neutrophil apoptosis under physiological in vivo conditions (page 10). Additionally, we measured GSDME cleavage in neutrophils isolated from infected mice and verified that GSDME cleavage occurs in vivo under inflammatory conditions (Fig.3e, described on page 8). Of note, under inflammatory conditions in vivo, dead neutrophils and extracellular GSDME were rapidly cleared. Consequently, the amount of cleaved GSDME in intact neutrophils collected from the inflamed site was expectedly lower than that detected in cultured neutrophils.
 2. **GSDME modulates the immune response by regulating efferocytosis.** We examined and confirmed this finding in multiple animal models. Disrupting GSDME enhanced the engulfment of dying neutrophils by macrophages (efferocytosis) in vivo, which in turn alleviated pulmonary inflammation. (Fig.4a-f, and Fig.6)
- In the revised manuscript, we also added a model of in vivo bacterial infection (Fig.7). Given our focus on inflammation and tissue damage within the lungs, we selected a *Staphylococcus aureus* (gram-positive bacteria)-induced pneumonia model. The newly obtained results not only provide insights into **the manner in which neutrophil GSDME modulates anti-bacterial functions** but also demonstrate that immune modulation mediated by GSDME is a general mechanism extending beyond LPS-elicited inflammation (page 12).

7. Furthermore, considering the availability of a Mrp8-cre x Gsdme fl/fl mouse system, it would be advantageous to replace the data obtained from the whole-body knockout system with the conditional knockout system.

Response: In the revised manuscript, all findings from the whole-body knockout have been validated using conditional knockout mice (Fig.5h-n, Fig.7, Fig.8, Fig.S12).

Reviewer #3

1. The most important finding is that neutrophils undergo spontaneous lytic death (pyroptosis) by spontaneous cleavage of GSDME. However, it is generally believed that apoptosis is a typical form of RCD in spontaneous neutrophil death and apoptotic neutrophils are cleared by efferocytosis. Therefore, this reviewer wonders whether spontaneous pyroptosis could be artificial and really occur in physiological in vivo situations. The authors should provide the data on spontaneous pyroptotic neutrophils (with GSDME cleavage) escaping from efferocytosis in mice.

Response:

- People have already started to appreciate the fact that neutrophil death is a heterogeneous process which includes both apoptosis and lytic cell death. The in vitro data are considered solid and convincing. Pyroptotic neutrophil death has been reported by multiple labs.
- As suggested by this reviewer, we added an experiment to examine the nature of neutrophil death in vivo (Fig.4k). Of note, the cells undergoing lytic death are fragile and may be destroyed during sample preparation. In addition, the apoptotic neutrophils are continuously engulfed by macrophages. Thus, the quantification may not accurately reflect the number of dying neutrophils at the inflamed site. However, we

indeed detected both apoptotic and pyroptotic neutrophils. The increase of neutrophil apoptosis elicited by GSDME disruption was also confirmed in this physiological in vivo situation (page 10).

- We also measured GSDME cleavage in neutrophils isolated from infected mice and confirmed that GSDME cleavage did occur in vivo under inflammatory conditions (Fig.3e, described on page 8).
- Additionally, to simulate neutrophil death during inflammation under pathophysiological conditions, we analyzed neutrophil death in the presence of various pro-inflammatory cytokines. Similar to what was observed during neutrophil spontaneous death, GSDME-deficient neutrophils under all tested conditions displayed a number of healthy cells comparable to that of the WT neutrophils. However, there was a noticeable shift towards apoptosis in the death mechanism of the former (Fig.S6, described on pages 6-7).

2. Another main concern is that the findings of this study are obtained only in mice, but not humans. Therefore, the translational value of the reported findings is difficult to assess. The authors should investigate the occurrence of spontaneous lytic neutrophil death and the role of GSDME in its process.

Response: Thanks for the comment. Reviewer #1 made a similar recommendation. We conducted the experiment as suggested and observed that GSDME was indeed cleaved, and the PR3> caspase-3> GSDME> pyroptotic cascade was operative in human neutrophils. These findings have been incorporated into Fig.S8 of the revised manuscript (described on page 8).

3. The authors showed that spontaneous apoptosis was promoted in GSDME-deficient neutrophils by living cell image and flow cytometry analyses. However, several inconsistent data between these analyses make obscure the conclusion. Although the percentage of apoptotic cells evaluated by AV+PI- cells was increased in GSDME-deficient neutrophils in early time points, it was unchanged in later time points. Furthermore, AV+PI+ cells including pyroptosis and secondary necrosis were increased or unchanged in GSDME-deficient neutrophils or unchanged. These results might reflect that GSDME deficiency enhanced caspase-3 activation and accelerated apoptosis in early time points. The present results are not sufficient to conclude that GSDME deficiency skews cell death from lytic to apoptotic cells.

Response:

- It is a great observation. Because many dead neutrophils disappear (turning into debris) during the process, the most accurate measurement for neutrophil death is the absolute number of neutrophils at each time point, rather than the percentage. Indeed, the results of living cell imaging and flow cytometry differ from each other. A proportion of swollen or lytic cells, detected by microscopy, will be lost during pipetting and processing for flow cytometry (Y. Teng, H. R. Luo, H. Kambara, Heterogeneity of neutrophil spontaneous death. Am J Hematol 92, E156-E159, 2017). Therefore, to accurately measure the absolute numbers of cell populations in FACS analysis, we routinely include flow cytometry beads in each FACS.
- Cell death analyzed by flow cytometry was largely consistent with that seen by microscopy. The absolute number of apoptotic cells was significantly higher in the GSDME-deficient neutrophil population, although the percentage was unchanged (Figure 2b-c). Additionally, although the percentage of healthy neutrophils (AV/PI double-negative) was lower in the *Gsdme* KO group than the WT group (Fig.2b), the absolute number of healthy cells was not affected by *Gsdme* knockout (Fig.2c).
- Our conclusion, that GSDME deficiency skews cell death from lytic to apoptotic cells, is based on the following results.

The live imaging, which can identify neutrophils undergoing apoptosis and lytic cell death, shows:

- The total number of healthy neutrophils remained unaltered.
- The absolute number of apoptotic neutrophils increased.
- The absolute number of pyroptotic neutrophils decreased.

The FACS analysis confirmed that:

- The total number of healthy neutrophils remained unaltered.
 - The ratio of apoptotic to pyroptotic neutrophils significantly increased.
- We have clarified and further discussed these points/results in the revised manuscript sample (discussed on pages 5-6).

4. Although the authors discussed the distinct functions of GSDMD and GSDME in neutrophil spontaneous death, the role of GSDMD was not fully assessed. Whereas the number of AV-PI- live cells was increased in GSDMD-deficient neutrophils, the number of dead cells was not shown. The number of AV+PI- neutrophils and AV+PI+ neutrophils must be increased in GSDMD-deficient neutrophils. Furthermore, the cleavage of GSDMD during neutrophil spontaneous death should be analyzed.

Response:

- The role of GSDMD in neutrophil death has been reported in multiple studies. We have elaborated on these findings in the revised manuscript (pages 7 and 15-17).
- Following the recommendation, we investigated and confirmed the cleavage of GSDMD during spontaneous neutrophil death (Fig.S7a, described on page 7).
- As indicated by the reviewer, the counts of AV+PI- and AV+PI+ neutrophils indeed decreased when GSDMD was disrupted. We have incorporated these results in Fig.S7c-d (described on page 7).

Technical concerns:

5. Although the authors used PI staining to evaluate the dead cells in live cell imaging, PI can be entered in living cells in a longer culture period. Other cell-impermeable dyes, such as SYTOX Orange/DeepRed should be tested.

Response: Thanks for the suggestion. We added an experiment in which SYTOX Orange was used to evaluate cell death (Fig.S3, described on page 5).

6. Some images of lytic cell death (Fig. 1A and B) are different from typical images of pyroptosis (swollen membrane and condensed nuclei). Morphological analysis should be validated at a fixed time point because culture under a microscope might affect viability.

Response: Thank you for the suggestion. We performed a morphological analysis at a fixed time point, without continuous monitoring under the microscope. The results can be found in Fig.S3. Additionally, we have expanded upon and discussed the morphological changes observed in pyroptotic neutrophils (e.g. swollen membrane and condensed nuclei) in the revised manuscript (Page 15).

7. The authors evaluated pyroptosis or lytic cell death by morphological change and PI staining. Other methods such as LDH release should be tested.

Response: We added the LDH release assay in the revised manuscript (Fig.1h, described on page 6).

REVIEWERS' COMMENTS

Reviewer #1 (Remarks to the Author):

The authors have been thorough and thoughtful in responding to the criticisms and recommendations raised by the three reviewers of the initial submission. This has included both new experiments (e.g. Figure 7) and expansion/ clarification of previous experiments.

The MS describes physiologically significant and mechanistically novel roles for neutrophil Gasdermin E (GSDME) in regulating the mode of neutrophil death between pro-inflammatory lytic pyroptosis and anti-inflammatory apoptosis. The study design incorporates both ex vivo experiments with isolated primary murine neutrophils (bone marrow or elicited peritoneal), human blood neutrophils, and four distinct in vivo murine models of neutrophil-mediated inflammation.

Reviewer #2 (Remarks to the Author):

The authors have demonstrated a thorough and thoughtful approach in addressing the raised questions. Especially, the validation of findings using conditional knockout mice addresses the initial concern about the whole-body knockout system, strengthening the robustness of the study.

Reviewer #3 (Remarks to the Author):

The authors responded adequately to the reviewer's comments, and this reviewer has no further comments for this paper.